# Large contributions of biogenic and anthropogenic sources to fine organic aerosols in Tianjin, North China

Yanbing Fan[1], Cong-Qiang Liu[1], Linjie Li[2], Lujie Ren[1], Hong Ren[1], Zhimin Zhang[1], Qinkai Li[1], Shuang Wang[1], Wei Hu[1], Junjun Deng[1], Libin Wu[1], Shujun Zhong[1], Yue Zhao[1], Chandra Mouli Pavuluri[1], Xiaodong Li[1], Xiaole Pan[2], Yele Sun[2], Zifa Wang[2], Kimitaka Kawamura[3], Zongbo Shi[4,1], and Pingqing Fu[1]

[1] Institute of Surface-Earth System Science, Tianjin University, Tianjin, 300072, China
[2] State Key Laboratory of Atmospheric Boundary Layer Physics and Atmospheric Chemistry, Institute of Atmospheric Physics, Chinese Academy of Sciences, Beijing, 100029, China
[3] Chubu Institute for Advanced Studies, Chubu University, Kasugai 487-8501, Japan
[4] School of Geography Earth and Environmental Sciences, University of Birmingham, Birmingham B15 2TT, United Kingdom

*Correspondence to*: Cong-Qiang Liu (liucongqiang@tju.edu.cn); Pingqing Fu (fupingqing@tju.edu.cn)

**Abstract.** In order to better understand the molecular composition and sources of organic aerosols in Tianjin, a coastal megacity in North China, ambient fine aerosol ($PM_{2.5}$) samples were collected on a day/night basis during November – December 2016 and May – June 2017. Organic molecular compositions in $PM_{2.5}$, including aliphatic lipids (*n*-alkanes, fatty acids and fatty alcohols), sugar compounds and photooxidation products from isoprene, monoterpene, β-caryophyllene, naphthalene and toluene, were analysed using gas chromatography-mass spectrometry. Fatty acids, fatty alcohols and saccharides were identified as the most abundant organic compound classes among all the tracers detected in this study during both seasons. High concentrations of most organics at night in winter may be attributed to intensive residential activities such as house heating and the low boundary layer height. Based on the tracer methods, the contributions of the sum of primary and secondary organic carbon (POC and SOC) to aerosol organic carbon (OC) were 24.8% (daytime) versus 27.6% (nighttime) in winter and 38.9% (daytime) versus 32.5% (nighttime) in summer. In detail, POC derived from fungal spores, plant debris, and biomass burning accounted for 2.78-31.6% (12.4%) of OC in the daytime versus 4.72-45.9% (16.3%) at night in winter, and 1.28-9.89% (5.24%) versus 2.08-47.2% (10.6%) in summer. Biomass burning derived OC was the predominant source of POC in this study, especially at night (16.0 ± 6.88% in winter and 9.62 ± 8.73% in summer). Biogenic SOC from isoprene, α/β-pinene and β-caryophyllene exhibited obvious seasonal and diurnal patterns, contributing 2.23 ± 1.27% (2.30 ± 1.35% in the daytime and 2.18 ± 1.19% at night) and 8.60 ± 4.02% (8.98 ± 3.67% and 8.21 ± 4.39%) to OC in winter and summer, respectively. Isoprene and α/β-pinene SOC were obviously elevated in summer, especially in the daytime, mainly due to strong photooxidation. Anthropogenic SOC from toluene and naphthalene oxidation contributed higher to OC in summer (21.0 ± 18.5%) than in winter (9.58 ± 3.68%). In summer, toluene SOC was the dominant contributor to aerosol OC, and biomass burning OC also accounted for a large portion to OC, especially in the nighttime,

which indicate that land/sea breezes also play an important role in aerosol chemistry at the coastal city of Tianjin in North China.

# 1 Introduction

The rapid industrialization in China has brought serious air pollution problem, with fine aerosol (PM$_{2.5}$, particles with diameters less than or equal to 2.5 μm) concentrations exceeding the standard in many regions. In particular, the North China Plain (NCP), the Yangtze River Delta (YRD) and the Pearl River Delta (PRD), where the economic development are at a leading level in China have been suffered from severe air pollution. In the past decade, atmospheric aerosols have been widely regarded as the major air pollutants in Chinese megacities (Chan and Yao, 2008; Aalto et al., 2001; Yang et al., 2016; Sun et al., 2018). In the lower troposphere, organic aerosols (OAs) account for about 20-90% of fine aerosols (Jimenez et al., 2009; Kanakidou et al., 2005; Zhang et al., 2007b). The scattering and absorption characteristics of OAs have great influences on regional atmospheric chemistry and radiation forcing. In addition, OAs can interference with cloud droplet nucleation and ozone formation through breaking the earth's radiation balance, which may further cause significant climate forcing (Ghan and Schwartz, 2007). It can also reduce visibility due to hygroscopicity and threaten human health, causing asthma, bronchitis, heart disease, cancer and other diseases (Pope et al., 2009). All adverse effects mentioned above are closely related to molecular composition and abundance of atmospheric organic aerosols (Kanakidou et al., 2005). Although organic aerosols in urban and rural regions (Simoneit et al., 1991b; Yang et al., 2016; Li et al., 2018), forests (Alves et al., 2001), mountain (Fu et al., 2008), islands (Zhu et al., 2015b; Zheng et al., 2018), coastal area (Feng et al., 2007; Kang et al., 2017) and remote oceans (Fu et al., 2011; Ding et al., 2013; Fu et al., 2013) have been studied based on identification by gas chromatography-mass spectrometry (GC-MS), the comprehensive and profound understanding of OAs in fine aerosols are still limited because of inadequate data on air pollution study in East Asia.

The Asian continent is an important source region of atmospheric aerosols that are emitted from biomass burning (BB), dust storms, fossil fuel combustion, as well as those formed through the photooxidation of biogenic and anthropogenic volatile organic compounds (VOCs). The NCP is considered to be one of the areas with the largest amount of biomass burning and anthropogenic emissions in the world (Andreae and Rosenfeld, 2008). Tianjin (39°N and 117°E), the largest coastal city of the NCP, located along the Haihe River and being adjacent to the Bohai Sea and East China Sea (Fig. 1), has suffered severe haze pollution along with rapid economic and industrial developments during the past decades. Fine particulate matters in the Tianjin atmosphere has high levels, which is urgently needed to investigate the chemical compositions and seasonal variations in organic molecules.

At present, a few studies have investigated the sources of atmospheric aerosols in Tianjin by analysing ionic species, heavy metals and organic and elemental carbon (Ho et al., 2012; Dong et al., 2013; Wang et al., 2015). Some studies pointed that secondary pollution, fossil fuel combustion, soil and construction dust are the main sources of PM$_{2.5}$ in Tianjin based on chemical mass balance models (Li et al., 2010; Wei et al., 2012). Xu et al. (2019) reported that coal combustion, secondary

inorganic aerosols, vehicle emissions, soil/road dust and industrial emissions contributed 10.9%, 44.4%, 16.1%, 13.1% and 9.7% to PM$_{2.5}$ in Tianjin during 2013–2016, respectively. In Tianjin aerosols collected during 2016 to 2017, the contributions of OC and EC to PM$_{2.5}$ was 17.5 ± 13.5% and 4.6 ± 3.6%, and the wind directions were dominated by southerly wind, which could bring more humid marine air masses to Tianjin (Ji et al., 2019). However, there have been few studies on organic aerosols in Tianjin at the molecular level. Tianjin is a typical coastal city, where organic aerosols may be influenced by both terrestrial and marine sources under the influence of land/sea breezes (Ding et al., 2004). Therefore, it is necessary to study the molecular compositions of atmospheric organic aerosols in Tianjin, which will be important for understanding the pollution characteristics and sources of atmosphere organic aerosols in coastal megacities.

In this study, we collected fine aerosol samples in urban Tianjin during the winter of 2016 and the summer of 2017. Ten organic compound classes (79 organic species) were identified, including aliphatic lipids, sugars compounds, biogenic and anthropogenic secondary organic aerosol (SOA) tracers. To better understand the primary emission sources and photo-oxidation formation, the contributions of different sources to organic aerosols in Tianjin were evaluated by tracer-based methods. The diurnal variations in organic aerosols under the apparent influence of sea and land breeze circulation were also discussed. Our findings are expected to enrich the database on the chemical characterization of organic aerosols in East China.

## 2 Experiments and Methods

### 2.1 Sample collection

Wintertime sampling was performed on the rooftop (approximately 20 m above ground level) of a teaching building on the Weijinlu Campus of Tianjin University (117.17°E, 39.11°N) in urban Tianjin (Fig. 1) during 10 November to 23 December 2016. Daytime sampling started from 8:00 to 20:00, while nighttime sampling from 20:00 to 8:00. Summertime sampling was performed from 22 May to 22 June 2017 (7:00 to 19:00 for daytime and 19:00 to 7:00 for nighttime). A high-volume air sampler (Tisch TE-PM2.5HVP-BL) was used for sampling at a flow rate of 1.0 m$^3$ min$^{-1}$. Aerosol samples were collected onto quartz fibre filters (Pallflex 2500QAT-UP), which were precombusted (450°C, 6h) to remove potential contamination of organics. Field blank filters were also collected in both seasons. After collection, the samples were wrapped by precombusted aluminium foils and were stored in darkness at -20°C until analysis. In total, 85 and 60 samples were collected in winter and summer, respectively.

### 2.2 Sample extraction and derivatization

A portion of each filter sample with the diameter of 24 mm was cut and ultrasonically extracted with chloromethane/methanol (2:1, v/v) for 10 min, which were repeated three times at room temperature. Quartz wool packed in Pasteur pipette was employed to filter the solvent extracts, and then concentrated by a rotary evaporator as well as dried

using pure nitrogen gas. The mixture with 50 μl of N,O-*bis*-(trimethylsilyl) trifluoroacetamide (BSTFA) and 1% trimethylsilyl chloride containing 10 μl of pyridine at 70°C for 3 h, afterwards were added and reacted with the extracts in order to make polar groups (e.g. COOH and OH) to be derivatized into the corresponding trimethylsilyl (TMS) esters and ethers (Schauer et al., 1996; Simoneit et al., 2004c; Fu et al., 2008). Finally, 40 μl of *n*-hexane containing internal standards

($C_{13}$ *n*-alkanes, 1.43 ng μl$^{-1}$) were added before gas chromatography-mass spectrometry (GC-MS) analysis. Field and laboratory blank filters were treated as real samples and applied for quality assurance and quality control.

## 2.3 Gas chromatography - mass spectrometry determination

Agilent model 7890A GC equipped with a 5975C mass-selective detector was applied to identify and quantified organic compound classes. There are split/splitless injector and DB-5MS fused silica capillary column, which is 30 m × 0.25 mm i.d

and 0.5 μm film thickness. The samples in the fused silica capillary column would be comply with the GC temperature program that 50°C remaining 2 min and increasing to 120°C at 15°C min$^{-1}$, afterwards to 300°C at 5°C min$^{-1}$, and finally held at 300°C last for 16 min. The carrier gas was helium. The MS detection was operated on the Electron Ionization (EI) mode at 70 eV, scanning from 50 to 650 Da. Most of the recoveries for authentic standards or surrogates were over 80%. The quality and quantity of single compound was acquired using the ChemStation software. Moreover, authentic standards were

employed to achieve GC/MS response factors. The results in our work had been corrected for the field blanks while not for recoveries.

## 2.4 OC and EC determination

Concentrations of OC and EC were measured by a thermal/optical carbon analyzer (model RT-4, Sunset Laboratory Inc., USA). The analytical errors were detected within ±10% through a duplicate analysis of each filter. In winter, the blank

levels were in the ranges of 1.52-2.84 μgC and 0-0.03 μgC for OC and EC, respectively. The summertime ranges of blank levels were 1.17-1.50 μgC and 0 μgC for OC and EC, respectively.

## 2.5 Calculated methods of POC and SOC contributions

In this study, the contributions of POC and SOC to total OC were evaluated based on organic tracer method. An experimentally derived factor of 13 pg C per spore was used to calculate the contribution of fungal spore to OC (Bauer et al.,

2008). Plant debris was evaluated by the tracer of cellulose and experimentally factor (Hans and Monika., 2003). The tracer mass fraction ($f_{SOC}$) factors 0.155 ± 0.039, 0.231 ± 0.111 and 0.023 ± 0.005 for isoprene, α-pinene and β-caryophyllene, respectively, were applied to evaluate the contributions of BSOAs to OC (Kleindienst et al., 2007).

# 3 Results and discussion

## 3.1 Meteorological conditions and air quality

Air quality data including AQI, PM$_{2.5}$ and quality grade are shown in Table S7 in the supporting information, which were available from the website of China air quality online monitoring and analysis platform. The variations in meteorological conditions and the concentrations of ambient PM$_{2.5}$ during the sampling periods are presented in Figure 2. Atmospheric pressure (P), temperature (T) and relative humidity (RH) fluctuated obviously in the winter of 2016, while they were relatively stable in the summer of 2017. The ambient T ranged from -3.26 °C to 11.8 °C (5.05 °C) in the daytime and from -3.56 °C to 9.12 °C (3.53 °C) at night in winter, while they were 17.3-34.8 °C (27.8 °C) in the daytime versus 16.0-30.4 °C (23.4 °C) at night in summer. The average P were 1023 ± 6.23 hPa in the daytime versus 1023 ± 5.51 hPa at night in winter and 1004 ± 4.24 hPa in the daytime versus 1004 ± 4.18 hPa at night in summer. The wintertime RH were in the ranges of 19.6-89.4% (53.8%) at daytime versus 34.5-96.4% (60.9%) at night, while the summertime values were 17.4-83.7% (39.0%) in the daytime versus 24.8-83.9% (50.2%) at night. The levels of PM$_{2.5}$ during sampling periods were in the ranges of 15-290 μg m$^{-3}$ (124 μg m$^{-3}$), much higher than summertime ones, with values as 12-73 μg m$^{-3}$ (42.9 μg m$^{-3}$). It is interesting to note that the diurnal variations in RH were similar with the pattern of PM$_{2.5}$, especially in winter, which may be attributed to the large percentage of secondary inorganic aerosols (SNA, including SO$_4^{2-}$, NO$_3^-$ and NH$_4^+$) in PM$_{2.5}$ (Tao et al., 2017). SNA were hygroscopic components that can cause quick heterogenic reactions under high RH conditions (Zheng et al., 2015; Xu et al., 2017). The levels of PM$_{2.5}$ showed high concentrations on November 30, December 4, 12, 18, 20 in 2016 and May 27-28, June 14, 18 in 2017 (Figure 2), relative to the sampling concentrations in the surrounding days. The wind direction (WD) at the sampling site was mainly south and southeast winds. There were four rainfall events occurred on November 20-22 in 2016, May 21-22 and June 5-6, 20-22 in 2017 during sampling periods. The concentrations of PM$_{2.5}$ decreased dramatically during the rain events.

## 3.2 Molecular compositions of fine organic aerosols and seasonal variations

### 3.2.1 *n*-Alkanes

The abundances and seasonal variations in *n*-alkanes during two seasons are shown in Fig. 3a. Concentrations of *n*-alkanes (C$_{18}$-C$_{35}$) were 343 ± 227 ng m$^{-3}$ (daytime) versus 499 ± 307 ng m$^{-3}$ (nighttime) in winter, which were roughly 2-3 times higher than those in summer with the average loadings at 141 ng m$^{-3}$ during both day- and night-time. In general, the molecular distributions of *n*-alkanes for most samples are characterized by an odd carbon number predominance with a maximum at C$_{23}$ (Fig. 3b) in winter versus high values at C$_{27}$ and C$_{29}$ in summer (Fig. 3c). The carbon preference index (CPI, concentration ratios of odd-carbon to even-carbon *n*-alkanes) ratios for C$_{18}$-C$_{35}$ *n*-alkanes of all samples were calculated, which is often used to identify the contributions of anthropogenic and biogenic sources (Simoneit, 1986). In winter, the CPIs were 1.21 ± 0.11 in the daytime and 1.19 ± 0.09 at night, which were comparable to those reported in Beijing and other urban aerosols from China (1.0 ± 0.43) (Wang et al., 2006). The average CPIs were 1.39 ± 0.40 in the daytime and 1.36 ± 0.44 at night in summer. The CPIs of terrestrial higher plant waxes are usually ~5-10, while CPIs close to unity are attributed

to marine sources and/or petroleum residues (Simoneit et al., 1991a; Hsu et al., 2006). Such molecular distributions indicate that aerosols in Tianjin may be mainly derived from incomplete combustion of fossil fuels/petroleum residue and/or marine sources tend to have similar CPIs in both seasons.

High molecular weight $n$-alkanes (HMW$_{alk}$, $C_{25}$-$C_{36}$) are mainly derived from terrestrial higher plant waxes, in which $C_{27}$, $C_{29}$

and $C_{31}$ are the dominant species. Low molecular weight $n$-alkanes (LMW$_{alk}$, < $C_{25}$) are usually emitted from biomass burning and fossil fuel combustion (Kawamura et al., 2003b; Freeman and Collarusso, 2001). The ratios of low molecular weight to high molecular weight (LMW/HMW$_{alk}$) were $1.03 \pm 0.30$ in the daytime versus $1.04 \pm 0.32$ at night in winter and $0.83 \pm 0.50$ in the daytime versus $0.56 \pm 0.32$ at night in summer, which indicate more important contributions of biomass burning and fossil fuels combustion in winter than those in summer. In winter, the concentrations of both LMW$_{alk}$ and

HMW$_{alk}$ at night were higher than those in the daytime (Fig. 3b and Table S1), which may be related to the enhanced anthropogenic activities (e.g. house heating) and the low boundary layer height at night. However, the concentrations of LMW$_{alk}$ were at higher levels in the daytime ($39.1 \pm 14.1$ ng m$^{-3}$) than that at night ($34.9 \pm 23.3$ ng m$^{-3}$) in summer (Fig. 3c and Table S1). Such distribution might be due to the significant sea and land breeze circulation, which could bring large amount of terrestrial higher plant waxes to Tianjin at night, while transport marine organic matters to mainland in the

daytime. The % of Wax $C_n$ is the contribution of biogenic $n$-alkane that is derived from high plant waxes (Ren et al., 2016). On average, the plant wax $n$-alkanes (WNA, $C_{25}$-$C_{34}$) accounted for 10.1% of total homologs in the daytime and 9.01% at night in winter, similar to those in summer with average contributions of 10.2% in the daytime and 9.91% at night (Table S2), indicating that higher plant waxes made a minor and stable contribution to $n$-alkanes in both seasons.

### 3.2.2 $n$-Fatty acids

The atmospheric abundances of and seasonal variations in fatty acids ($C_{12}$-$C_{32}$) in Tianjin aerosol samples are shown in Fig. 4a and Table S1, including two unsaturated fatty acids (palmitoleic acid ($C_{16:1}$) and oleic acid ($C_{18:1}$)). Molecular distributions of saturated fatty acids showed a strong even carbon number predominance with two maxima at $C_{16:0}$ and $C_{18:0}$ both in winter and summer (Fig. 4b-c). Such a pattern is similar to other urban aerosols in China (Zhao et al., 2014), India (Fu et al., 2010b) and USA (Schauer et al., 2002), and mountain aerosols (Kawamura et al., 2003b; Fu et al., 2011). The CPI (concentration

ratios of even-carbon over odd-carbon for $C_{20}$-$C_{30}$ fatty acids) values ranged from 1.52-8.59 (3.09) in the daytime and those were in the range of 0.29-4.11 (on average 2.53) at night in summer (Table S2). The average CPI values were similar to the daytime (2.96) and nighttime (2.62) averages in winter. Moreover, the CPIs of two seasons were slightly lower than to comparable with marine aerosols over the Arctic Ocean (1.9-8.0, 4.4), indicating that biogenic emissions made important contributions in both seasons (Fu et al., 2013b).

In this study, the total concentrations of both saturated and unsaturated fatty acids were $666 \pm 418$ ng m$^{-3}$ (daytime) versus $778 \pm 448$ ng m$^{-3}$ (nighttime) in winter, and $410 \pm 354$ ng m$^{-3}$ (daytime) versus $387 \pm 340$ ng m$^{-3}$ (nighttime) in summer. High molecular weight fatty acids (HMW$_{fat}$, $\geq C_{20:0}$) are mainly derived from terrestrial higher plant waxes (Kawamura et al., 2003a), and low molecular weight fatty acids (LMW$_{fat}$, $\leq C_{19:0}$) have multiple sources such as vascular plants, microbes, cooking emissions, and marine phytoplankton (Fu et al., 2008a;Cox et al., 1982). Biomass burning is also a source of fatty

acids (Zhang et al., 2007a; Fu et al., 2012). In winter, the overall concentrations of saturated $LMW_{fat}$ and $HMW_{fat}$ were $442 \pm 353$ ng m$^{-3}$ and $191 \pm 132$ ng m$^{-3}$ in the daytime, lower than those ($477 \pm 283$ ng m$^{-3}$ and $234 \pm 156$ ng m$^{-3}$) at night. During the summertime period, the concentration of $HMW_{fat}$ (3.98-64.6 ng m$^{-3}$, 27.0 ng m$^{-3}$) in the daytime was lower than that (0.68-198 ng m$^{-3}$, 35.9 ng m$^{-3}$) at night, while the daytime concentrations of $LMW_{fat}$ (51.0-1260 ng m$^{-3}$, 366 ng m$^{-3}$) affected

by sea and land breeze circulation, which were higher than those (77.0-1556 ng m$^{-3}$, 332 ng m$^{-3}$) at night. So, the average ratios of LMW/HMW fatty acids are used to evaluate the relative contributions of terrestrial and marine sources to the abundance of fatty acids in ambient aerosols in Tianjin (Table S2). In winter, the average ratios of saturated LMW/HMW fatty acids were 3.35 (daytime) and 2.77 (nighttime), while the ratios were much higher in summer as 15.2 (daytime) and 19.0 (nighttime). Such patterns suggest that the aerosols were largely influenced by marine air masses in summer, tending to

have higher LMW/HMW ratios, whereas the distributions in winter were possibly associated with enhanced anthropogenic activities (e.g. biomass burning) from mainland and the low boundary layer height at night.

Unsaturated fatty acids are reported to be directly emitted from multiple sources such as leaf surfaces of plants (Rogge et al., 1993), wood combustion (Fine et al., 2001), meat charbroiling (Nolte et al., 1999) and marine biota (Fu et al., 2013;Kawamura and Gagosian, 1987). Moreover, unsaturated fatty acids can be rapidly oxidized by ozone, $H_2O_2$ or OH

radicals (Kawamura and Gagosian, 1987), therefore they can be used to study the reactivity and aging processes of atmospheric aerosols (Rudich et al., 2007). In winter, the average ratios of $(C_{16:1}+C_{18:1})/(C_{16:0}+C_{18:0})$ were 0.10 (daytime) versus 0.28 (nighttime) in winter, and 0.08 versus 0.20 in summer. The low daytime levels in both seasons suggest that unsaturated fatty acids have undergone photochemical degradation in the daytime, which also implies that the secondary organic aerosols (SOAs) maybe ubiquitous during both winter- and summer-time in Tianjin. What need to focus on is that

$C_{16:0}$ and $C_{18:0}$ were two species with the most abundant concentrations and $C_{18:0}/C_{16:0}$ is a useful tool for source identification of fatty acids. The ratios lower than 0.25 indicate that fatty acids are mainly derived from foliar vegetation combustion, waxy leaf surface abrasions and wood smoke. The ratios ranging from 0.25 to 0.5 refer that fatty acids may come from the exhausts of car and diesel truck, and those in the range of 0.5-1.0 suggest the cooking emission and paved or unpaved road dust make a contribution to fatty acids (Rogge et al., 2006). In winter, the ratios of $C_{18:0}/C_{16:0}$ in aerosols were about 0.55

both in day- and night-time, lower than the values in summer (0.77 versus 0.78 for day- and night-time), implying an intense input from anthropogenic activities such as incomplete combustion of fossil fuels and biomass burning in winter, while the fatty acids may be largely attributed to the emissions of cooking and/or vehicles as well as road dust in summer.

### 3.2.3 *n*-Alcohols

The abundances of and seasonal variations in normal fatty alcohols in organic aerosols are shown in Fig. 5a. *n*-Alcohols

($C_{12}$-$C_{31}$) were detected in the aerosol samples with concentrations of $1310 \pm 811$ ng m$^{-3}$ (daytime) versus $1520 \pm 1010$ ng m$^{-3}$ (nighttime) in winter and $621 \pm 367$ ng m$^{-3}$ (daytime) versus $572 \pm 438$ ng m$^{-3}$ (nighttime) in summer, being apparently higher than the concentrations of *n*-alkanes and fatty acids. The molecular distributions showed strong even-carbon numbered predominance with the predominance at $C_{16}$ and $C_{18}$ (Fig. 5b-c). High molecular weight ($HMW_{alc}$, > $C_{19}$) *n*-alcohols are mainly derived from higher plant waxes and biomass burning (Wang et al., 2006). Low molecular weight

(LMW$_{alc}$, C$_{12}$-C$_{19}$) *n*-alcohols are related to marine and soil microbes (Fu et al., 2008b). The total concentrations of LMW$_{alc}$ and HMW$_{alc}$ were in the ranges of 223-2830 ng m$^{-3}$ (850 ng m$^{-3}$) and 84.8-1400 ng m$^{-3}$ (455 ng m$^{-3}$) in the daytime, lower than those at night as 142-2910 ng m$^{-3}$ (972 ng m$^{-3}$) and 92.2-1730 ng m$^{-3}$ (549 ng m$^{-3}$) in winter, respectively. In contrast, the concentrations of LMW$_{alc}$ (89.0-1320 ng m$^{-3}$, 506 ng m$^{-3}$) and HMW$_{alc}$ (28.7-238 ng m$^{-3}$, 115 ng m$^{-3}$) in the daytime were higher than those (139-1627 ng m$^{-3}$, 467 ng m$^{-3}$ and 19.2-474 ng m$^{-3}$, 105 ng m$^{-3}$) at night in summer, respectively (Fig. 5a and Table S1). Compared with HMW$_{alc}$, the concentrations of LMW$_{alc}$ were significant higher in the daytime than those at night, which may in part due to the more significant sea breezes during the daytime and land breezes at night. Therefore, we infer that the fatty alcohols in winter may mainly be derived from biomass burning and soil resuspension particles, while the contributions of marine/biogenic emission and biomass burning can possibly explain the molecular distribution of fatty alcohols in summertime aerosols.

The relative abundances (%) of HMW$_{alk}$, HMW$_{fat}$ and HMW$_{alc}$ in the Tianjin aerosols are illustrated in a triangular diagram (Fig. 6). The average abundances of HMW$_{alc}$ were 54.5% (wintertime) and 44.6% (summertime), which were the most dominant species among aliphatic lipids. The percentage of HMW$_{alk}$ was lower in winter (22.5%) than that in summer (42.5%). The relative abundances of HMW$_{alk}$, HMW$_{fat}$ and HMW$_{alc}$ in the aerosols collected over the East China Sea (Kang et al., 2017), at Chichi-Jima Island in the northwest Pacific (Kawamura et al., 2003b), at Mt. Tai (Fu et al., 2008b), and urban Beijing (Ren et al., 2016) are plotted as category A, B, C and D. It is worthy to note that the results of this study overlapped with other four groups, which indicates the aerosols may have similar source contributions to some extent, and highlights that the coastal city Tianjin were influenced by the mixture of terrestrial and marine air masses.

It was found that most of the positions of wintertime data overlapped with aerosols collected from Mt. Tai (group C) and a fraction of samples fell in D area that represents the Beijing aerosols. Fu et al. (2008) reported that the field burning of wheat straws largely contributed to the Mt. Tai aerosols, which could be further transported to the Pacific Ocean under the influences of the westerly wind from mainland. The Beijing aerosols in group D were significantly affected by incomplete fossil fuel combustion and biomass burning as well as biogenic emissions (Ren et al., 2016). However, most of the summertime aerosols in Tianjin covered the group A, suggesting they may share similar sources with aerosols collected over the East China Sea (Kang et al., 2017), which may be significantly influenced by biogenic/marine emissions under the marine air masses and BB through long-range transport.

### 3.2.4 Molecular distributions of sugars and sugar alcohols

Fourteen sugar compounds including 3 anhydrosugars, 5 primary saccharides and 6 sugar alcohols identified in this study (Table S1) are water-soluble, which are known to contribute to water-soluble organic carbon (WSOC) (Graham et al., 2003; Elbert et la., 2007; Fu et al., 2008). In addition, they can affect the aerosol hygroscopicity (Mochida and Kawamura, 2004; Fu et al., 2008) and regulate climate to some extent (Kanakidou et al, 2005). The average concentrations of total sugars were 371 ± 208 ng m$^{-3}$ (daytime) versus 496 ± 247 ng m$^{-3}$ (nighttime) in winter and 61.2 ± 21.2 ng m$^{-3}$ (daytime) versus 96.9 ± 94.0 ng m$^{-3}$ (nighttime) in summer.

Levoglucosan, a dominant tracer of biomass burning (Graham et al., 2002; Sheesley et al., 2003; Hays et al., 2005; Iinuma et al., 2007; Fu et al., 2008), was the most abundant compound among anhydrosugars and total sugars as well, with average concentrations of $205 \pm 122$ ng m$^{-3}$ (daytime) versus $296 \pm 153$ ng m$^{-3}$ (nighttime) in winter and $12.8 \pm 6.97$ ng m$^{-3}$ (daytime) versus $34.4 \pm 46.0$ ng m$^{-3}$ (nighttime) in summer. The contribution of biomass burning was much more significant in winter, especially at night either in summer or winter. It has two isomers, namely galactosan and mannosan, which were detected in all aerosol samples (Fig. 7 and Table S1). These anhydrosugars are formed through the pyrolysis of cellulose/hemicellulose in different types of biomass such as grasses (Iinuma et al., 2007), woods (Graham et al., 2002) and agricultural residues including wheat (Fu et al., 2008b) and rice straws (Sheesley et al., 2003). These three anhydrosugars showed similar seasonal patterns (Fig. 7a-c) with the significantly higher wintertime concentrations than those in summer.

The ratios of levoglucosan to mannosan (L/M) and mannosan to galactosan (M/G) have been applied to discriminate different categories of biomass burning. Figure 8 shows the isomer ratios of crop residues, soft and hard wood from a number of regions in different countries reported in other literatures. All the values in previous studies were related to the PM$_{2.5}$ aerosol samples. The average ratios of L/M were reported in the ranges of 3.0-5.80, 12.9-35.4 and 40.0-41.6 for smoke emitted by the burning of softwood, hardwood and crop residues, respectively (Sheesley et al., 2003; Fine et al., 2004; Oros et al., 2006; Engling et al., 2006). Besides, the average ranges of M/G as 3.60-7.0, 1.2-2.0 and 0.30-0.60 respectively represented the burning of softwood, hardwood and crop residues (Sheesley et al., 2003; Fine et al., 2004; Oros et al., 2006; Engling et al., 2006). The relative contributions of levoglucosan and mannosan in aerosols can be used to indicate the apportionment of cellulose and hemicellulose in biomass fuels. The low M/G values suggest that the smokes may be mainly derived from the burning of biomass briquettes, crop straws and grasses (Sheesley et al., 2003; Oros et al., 2006; Fu et al., 2008). In this study, the L/M ratios (4.88-19.8, 7.38) were high and the M/G ratios (1.0-5.51, 1.59) were low in winter, while the ranges of summertime aerosols were detected as 2.74-15.8 (5.68) and 1.15-2.84 (2.12) for L/M and M/G, respectively. These results suggest that the soft- and hard-wood are the main types of biofuels in both seasons in Tianjin, and the use of hardwood and/or crop residues was enhanced in winter.

Primary saccharides (fructose, glucose, xylose, sucrose, trehalose) with substantially similar seasonal distributions (Fig. 7d-h) have been widely employed to indicate the sources of resuspension of surface soils and unpaved road dust containing biological materials (Simoneit et al., 2004a; Fu et al., 2008). The total concentrations of primary saccharides were $46.8 \pm 20.9$ ng m$^{-3}$ (daytime) versus $49.9 \pm 23.0$ ng m$^{-3}$ (nighttime) in winter and $20.1 \pm 11.1$ ng m$^{-3}$ (daytime) versus $21.2 \pm 21.5$ ng m$^{-3}$ (nighttime) in summer, showing no obvious diurnal variations. The results imply that the resuspension of surface soils and unpaved road dust containing biological materials was a quite stable source to organic aerosols in Tianjin. We also determined 6 sugar polyols, consisting of arabitol, mannitol, glycerol, erythritol, xylose and maltose. Glycerol was the most abundant sugar alcohols in both seasons. The average concentrations of glycerol were $49.8 \pm 32.0$ ng m$^{-3}$ (daytime) versus $52.8 \pm 27.1$ ng m$^{-3}$ (nighttime) in winter and $14.8 \pm 10.8$ ng m$^{-3}$ (daytime) versus $25.0 \pm 31.9$ ng m$^{-3}$ (nighttime) in summer. The levels of glycerol had a notable enrichment in winter, higher than the concentrations of primary saccharides, as well as were positively correlated with levoglucosan ($R^2 = 0.73$, $p < 0.01$, N = 85; Table S3), implying that there were potential

emissions from biomass burning for glycerol in winter. In summer, however, the low correlations ($R^2 = 0.32$, $\rho < 0.05$, N = 60; Table S4) founded between glycerol and levoglucosan suggest that most of the glycerol were possibly derived from the metabolism of soil microorganisms (Simoneit et al., 2004b; Li et al., 2018) as well as photooxidation. Moreover, arabitol and mannitol were detected with similar seasonal trends (Fig. 7i-j), representing the contributions of fungal spores (Bauer et al., 2008; Zhu et al., 2016) that are prevalent in land and marine areas (Elbert et al., 2007; Zhu et al., 2015).

In Figure 7, it is obvious that the concentrations of levoglucosan (a specific tracer of biomass burning), saccharides and sugar alcohols were more abundant in winter than in summer. In addition, saccharides and sugar alcohols had high levels especially when the levoglucosan reached peaks in both seasons, implying that they may share the similar sources to some extent. To verify the contribution of biomass burning to levoglucosan and sugar alcohols, we downloaded the fire maps in winter and summer during the sampling periods (Figure S1). It can be seen clearly that there were large amounts of anthropogenic activities in winter (Figure S1b). As we all know, the combustion of fossil fuels and biofuels were widely used for house heating in winter in China. The biomass burning could contribute to levoglucosan and sugar alcohols. In summer, there were relatively dense fire spots distributed in the North China Plain (NCP) and some south agricultural provinces, such as Jiangsu, Anhui, Henan and Shandong (Figure S1a). Fu et al. (2008) had reported that there was large-scale burning of wheat straw across the country during May-June. Meanwhile, the wind direction was mainly southerly during winter- and summertime in Tianjin (Figure 2). Therefore, in addition to the contribution of local biomass burning, the southern wind carried a large number of biomass burning particulate matters from NCP and southern agricultural provinces to Tianjin, which could be the source of levoglucosan and sugar alcohols.

In this study, we also calculated the contributions of different saccharides to total sugars. It was found that the average percentages of anhydrosugars to total sugars (Table S2) were $0.67 \pm 0.05$ (daytime) versus $0.73 \pm 0.05$ (nighttime) in winter, which were roughly 3 times higher than those ($0.27\pm0.08$ in the daytime versus $0.37 \pm 0.15$ at night) in summer. In contrast, the percentages of primary saccharides and two main tracers of fungal spores (arabitol and mannitol) with average ratios as $0.16 \pm 0.04$ (daytime) and $0.13 \pm 0.02$ (nighttime) in winter, which were about 3 times lower than those of summertime aerosols ($0.43 \pm 0.14$ in the daytime versus $0.30 \pm 0.11$ at night). Meanwhile, the contributions of anhydrosugars were high at night, especially in winter, while the large proportions of primary saccharides and sugar alcohols were identified in summer, especially for the daytime (Figure 9-10 and Table S2). These results suggest that biomass burning made a significant contribution in winter, accounting for 68.3% and 74.0% of total sugar sources for the day- and night-time, while the summertime aerosols were apparently influenced by biological sources, especially in the daytime. Primary saccharides and sugar polyols were responsible for 32.7% versus 40.0% of total sugars during the daytime and 22.8% versus 40.1% of total sugars during nighttime in summer, respectively (Fig. 10).

### 3.2.5 Biogenic SOA tracers

The biosphere-atmosphere hydrocarbon exchange is usually subject to the global biogenic emission with the isoprene loading of 600 Tg per year (Sharkey et al., 2007). The reactive double bonds of isoprene can be easily oxidized by oxidants such as OH, $NO_3$ and $O_3$ in the atmosphere. Six molecular markers were identified as isoprene SOA tracers, i.e., 2-methylglyceric acid (2-MGA), $C_5$-alkene triols (*cis*-2-methyl-1,3,4-trihydroxy-1-butene, *trans*-2-methyl-1,3,4-trihydroxy-1-butene and 3-methyl-2,3,4-tri-hydroxy-1-butene) and two 2-methyltetrols (MTLs, 2-methylthreitol and 2-methylerythritol) (Table S1). The wintertime concentrations of total isoprene SOA tracers were 1.03-10.6 ng m$^{-3}$ (4.13 ng m$^{-3}$) in the daytime versus 0.53-11.7 ng m$^{-3}$ (4.54 ng m$^{-3}$) at night, which were much lower than summertime samples with values of 3.63-83.9 ng m$^{-3}$ (29.6 ng m$^{-3}$) in the daytime and 5.40-106 ng m$^{-3}$ (25.3 ng m$^{-3}$) at night. In winter, 2-MGA was the most abundant species, followed by $C_5$-alkene triols and MTLs. However, the levels of 2-MGA were lower than concentrations of $C_5$-alkene triols and MTLs in summer.

The average concentrations of MTLs were detected as $0.43 \pm 1.17$ ng m$^{-3}$ (daytime) versus $0.40 \pm 1.03$ ng m$^{-3}$ (nighttime) in winter and $12.1 \pm 8.70$ ng m$^{-3}$ (daytime) versus $10.6 \pm 11.2$ ng m$^{-3}$ (nighttime) in summer. $C_5$-alkene triols are specific isoprene SOA tracers under low-$NO_x$ conditions (Surratt et al., 2010), accompanying similar seasonal variations with 2-methyltetrols in summer (Fig. 11b-c), while the 2-MGA is a further oxidation product of isoprene under high $NO_x$ conditions (Surratt et al., 2010). The concentrations of 2-MGA were $2.13 \pm 1.81$ ng m$^{-3}$ (daytime) versus $2.32 \pm 2.15$ ng m$^{-3}$ (nighttime) in winter and $5.76 \pm 3.89$ ng m$^{-3}$ (daytime) versus $4.27 \pm 3.98$ ng m$^{-3}$ (nighttime) in summer. In general, $C_5$-alkene triols are recognized as significant terrestrial tracers (Fu et al., 2014), which had a good correlation ($R^2 = 0.83$, $p < 0.01$, N = 60, Table S5) with MTLs in summer, suggesting that they may have the similar terrestrial sources such as biomass burning and higher plant waxes. However, there was no correlation in winter (Table S5), implying different sources of these two species in winter. Meanwhile, the average 2-MGA over MTLs (2-MGA/MTLs) ratios were 17.0 (daytime) versus 22.0 (nighttime) in winter, much higher than those (0.54 versus 0.50 for day- and night-time, respectively) in summer, and the higher values in winter are possibly due to the higher concentrations of $NO_x$ and acidity on SOA formation (Surratt et al., 2007). Which can also explain the phenomenon that the concentrations of 2-MGA were higher in winter than in summer.

Four oxidation products of α/β-pinene were detected in aerosol samples, including 3-hydroxyglutaric acid (3-HGA), 3-methyl-1,2,3-butanetricarboxylic acid (MBTCA), pinonic and pinic acids. The total average concentrations of α/β-pinene SOA tracers were $12.2 \pm 7.69$ ng m$^{-3}$ (daytime) versus $11.4 \pm 6.02$ ng m$^{-3}$ (nighttime) in winter and $23.4 \pm 13.6$ ng m$^{-3}$ (daytime) versus $22.4 \pm 23.7$ ng m$^{-3}$ (nighttime) in summer. Positive correlations were found between isoprene and α/β-pinene SOA tracers as well as T in summer (Table S6), while correlations in winter were weak, indicating they may have similar sources or influencing factors such as T and RH in summer and large amounts of different origins in winter.

Among the α/β-pinene SOA tracers, the most predominant compound was pinonic acid, followed by pinic acid. The concentration of 3-HGA was higher than MBTCA in winter, while it was opposite in summer (Table S1). Pinonic and pinic acids are the first-generation products of α/β-pinene oxidation, which can be further photo-degraded into products such as

MBTCA (Claeys et al., 2007). Therefore, the aging level of α/β-pinene could be evaluated using values of pinonic and pinic acids to MBTCA ((pinonic + pinic)/MBTCA) (Gómez-González et al., 2012; Ding et al., 2014). In our work, ratios of (pinonic + pinic)/MBTCA were 17.6 ± 15.9 (daytime) versus 31.0 ± 51.9 (nighttime) in winter, which were much higher than the summertime aerosols (6.8 ± 7.0 in the daytime versus 7.3 ± 9.7 in the nighttime), suggesting that the summertime aerosols were more aged than those in winter due to strong photo-oxidation in summer.

Isoprene emissions are more susceptible to higher temperature with larger contributions in summer (Li et al., 2018). The ratio of isoprene tracers to α/β-pinene tracers ($R_{iso/pine}$) can also be used to evaluate the relative contributions of isoprene and α/β-pinene oxidation to biogenic SOA formation. In the present study, the average $R_{iso/pine}$ ratios were 0.39 (daytime) versus 0.52 (nighttime) in winter, which were lower than summertime samples (1.85 in the daytime versus 1.77 in the nighttime), suggesting that isoprene oxidant products were more abundant in summer than those in winter. In addition, the ratios are higher than aerosols in Hong Kong (average 0.46) (Hu et al., 2008) during summertime, whereas lower than those of mountain aerosols such as Mt. Tai with the averages 4.9 and 6.7 for the day- and night-time in summer (Fu et al., 2010) and Mt. Changbai (3.7 in the daytime) (Wang et al., 2008). This is reasonable because mountain aerosols contain much more biogenic SOA than urban aerosols.

β-Caryophyllene is one of the most abundant sesquiterpene compounds reported frequently in previous studies due to its high reactivity and relatively low vapor pressure (Fu et al., 2010), which could be emitted from plants such as pine and birch trees (Helmig et al., 2006; Duhl et al., 2008). β-Caryophyllinic acid is an ozonolysis or photo-oxidation product of β-caryophyllene (Jaoui et al., 2007), which was detected with concentrations of 10.7 ± 9.33 ng m$^{-3}$ (daytime) versus 10.3 ± 8.41 ng m$^{-3}$ (nighttime) in winter, being five times higher than summertime aerosols (1.99 ± 1.81 ng m$^{-3}$ in the daytime and 2.21 ± 4.53 ng m$^{-3}$ at night). Previous studies in Okinawa (Zhu et al., 2016), India (Fu et al., 2010b) and Beijing (Li et al., 2018) also reported the concentrations of β-caryophyllinic acid maximized in winter. There are large amounts of sesquiterpenes attached to woods and leaves due to the low volatility, which could be emitted from smoke of the biomass burning, especially in winter (Zhu et al., 2016). The seasonal variation in the sesquiterpene SOA tracer was likely controlled by air masses from Southeast Asia, carrying more oxidized β-caryophyllinic acid via long range transport (Fu et al., 2010b; Zhu et al., 2016). It is interesting to note that β-caryophyllinic acid, levoglucosan (a specific tracer of biomass burning) and 2,3-dihydroxy-4-oxopentanoic acid (DHOPA) had similar seasonal variations (Fig. 7a and 11h), with a high concentration peak occurred on a severe haze episodes (May 28, 2017). Moreover, there were strong positive correlations between these species in both seasons (Table S5-6), which suggest that elevated biomass burning could attribute to β-caryophyllinic acid during the haze periods (Fig. 11h).

**3.3 Effects of meteorological condition to organic tracers**

Meteorological condition plays an important role in the temporal variations in organic aerosols. In winter, the wind directions (WD) were variable, with the predominance of south and southwest winds and the relative humidity (RH) also changed abruptly. Stagnant meteorological conditions caused frequent occurrence of haze events in the North China Plain. In

general, the concentrations of biogenic SOA tracers were higher during the daytime than nighttime because of the high emissions of biogenic VOCs followed by photooxidation in the daytime. Fu et al. (2016) also reported that the biogenic SOA tracers had higher levels in the daytime than nighttime both during winter- and summertime in Mumbai, India. In this study, the daytime concentrations of biogenic SOA tracers were also higher than that at night in summer, while universally lower than nighttime ones in winter (Fig. 11). Such results might be attributed to the sea and land breeze circulation in the coastal city of Tianjin. At night, land breezes from the Asian continent carry a large number of terrestrial/anthropogenic organic matters.

A haze episode (Ep1) was occurred on May 28 in 2017, when the wind direction changed from south-southwest to southeast (Fig. 2). Meanwhile, the levels of OC increased rapidly from 3.65 μg m$^{-3}$ up to 6.54 μg m$^{-3}$ in this day, and then decreased to its previous level due to a heavy rainfall event on May 29. It is interesting to note that most of the organic tracers, especially for biogenic SOA tracers reached higher levels in the nighttime than daytime (Fig. 11). Fu et al. (2008) reported that the south wind from provinces including Anhui, Jiangsu, Shandong, Henan and Hebei, where had experienced agricultural waste burning, may carry more emissions of wheat straw combustion products via long-range transport during May to June period in summer. In contrast, there was another haze episode (Ep2) occurred on June 19, when WD changed from southwest to southeast (Fig. 2). The significantly high concentrations of most biogenic SOA tracers in the daytime possibly because the sea breezes bring large amount of biogenic organic matters. In addition, the effects of temperature cannot be ignored in this work. For example, the concentrations of arabitol and mannitol peaked on June 9 (Fig. 7i-j), which was in line with the high temperature on that day, indicating the increasing temperature enhanced the biological activity. Likewise, the phthalate esters reached peaks as well, suggesting that elevated temperature promotes the evaporation of phthalate esters from plastic products (Fujii et al., 2003; Wang et al., 2006), which will be studied afterwards.

In this study, four rain events were recorded during the sampling periods. It is interesting to note that there were obvious differences between winter- and summer-time samples in terms of the contributions of primary and secondary OC to total OC on rainy and fine days (Figure 12). The concentrations of primary and secondary OC decreased dramatically on rainy days in both seasons, mainly due to the washout effect on pollutants. In winter, the levels of primary OC were higher than secondary OC (mainly from anthropogenic VOCs) before the rain events. Although the concentrations of primary and secondary OC decreased on rainy days, the level of primary OC had a substantial reduction (Figure 12a). However, in summer, the concentrations of secondary OC (both biogenic and anthropogenic SOC) were significantly higher than primary OC before the rain event. We found that the summertime rain event affected little on the levels of primary OC and biogenic SOC, but it decreased the anthropogenic SOC obviously. Such seasonal differences may be attributed to the important and persistent sources such as fossil fuel combustion and biomass burning in the local regions in winter and biogenic VOC emissions in summer.

### 3.4 Source apportionment based on organic molecular markers

### 3.4.1 Contributions of BB, fungal spores and plant debris to OC

OC concentrations detected in this study were in the ranges of 5.33-79.8 µg m$^{-3}$ (23.7 µg m$^{-3}$) in winter and 1.43-6.64 µg m$^{-3}$ (3.78 µg m$^{-3}$) in summer. The Asian summer monsoon may bring clean marine air masses to Tianjin and lower the atmospheric levels of OC in summer (Mao et al., 2004). In detail, the concentrations of BB-derived OC were the most abundant in winter (2.49 ± 1.48 µg m$^{-3}$ versus 3.61 ± 1.86 µg m$^{-3}$ for day- and night-time), which were significantly higher than those in summer (0.16 ± 0.08 µg m$^{-3}$ and 0.42 ± 0.56 µg m$^{-3}$) based on the ratio of levoglucosan to OC (L/OC) of 8.2% (Andreae and Merlet, 2001;Zhang et al., 2007a). However, the L/OC ratio was just one of the experimental simulation results, which we think it is reasonable in this study. And there are plenty of literatures to show that L/OC ratios are variable depending on the sources of biomass (e.g. straw and wood) and burning conditions. On average, BB contributed to OC as 12.1% (daytime) versus 16.0% (nighttime) in winter and 4.14% (daytime) versus 9.62% (nighttime) in summer, which were roughly comparable to the values in the range of 1.0-35% (9.9%) for a whole year in a nearby megacity of Beijing (Li et al., 2018).

Mannitol and arabitol are generally employed to evaluate the numbers of fungal spores (Elbert et al., 2007; Bauer et al., 2008). This work uses the values of 1.7 pg mannitol per spore and 13 pg OC per spore (Bauer et al., 2008) in order to identify the contributions of fungal spores-derived OC. Meanwhile, the plant debris-derived OC were estimated based on the relationship of glucose and plant debris as well as the OM/OC ratio of 1.93 (Puxbaum and Tenze-Kunit, 2003). Here, the estimations based on molecular tracers indicate that fungal spores and plant debris had higher contributions to OC during summertime than wintertime (Fig. 13a-b and 14) due to the high biological activities in summer. The contributions of fungal spore-derived OC were 0.88% (daytime) versus 0.79% (nighttime) in summer, which were lower than the values in Beijing (2.79% in summer) (Li et al., 2018) and much lower than the forest aerosols in Japan with levels of 22% in the daytime versus 45% at night (Zhu et al., 2016) and 12.1% in a tropical rainforest aerosols on Hainan Island (Zhang et al., 2010). Plant debris-derived OC accounted for the lowest proportion of primary OC with percentages accounting for primary OC at 0.22% for both the day- and night-time in summer, lower than summertime aerosols in Beijing (1.05%) (Li et al., 2018).

### 3.4.2 Contributions of biogenic and anthropogenic VOCs

SOAs formed from biogenic and anthropogenic precursors are important contributors in PM$_{2.5}$. It has been reported that global SOAs mostly originates from BVOCs based on the atmospheric modelling studies of SOA (Hallquist et al., 2009). Global BVOCs emissions can reach 760 TgC per year, among which isoprene, monoterpenes and sesquiterpenes account for 70%, 11% and 2.5%, respectively (Sindelarova et al., 2014). Anthropogenic SOCs from toluene and naphthalene were respectively estimated by 2,3-dihydroxy-4-oxopentanoic acid (DHOPA) and phthalic acid with mass fractions of 0.0026 and 0.0199 as well as the OM/OC value of 1.93 (Kleindienst et al., 2007; Fu et al., 2014b). Vehicle emissions may be the main sources for toluene in Tianjin, and solvent usage in electronics and chemical industries are also the potential sources

(Widiana et al., 2019; Fu et al., 2016). McFiggans et al. (2019) demonstrated that there would be a substantial overestimation of SOA production, because of the simple linear addition of SOA mass yields from the individual yields of components in a VOC mixture. The measurement uncertainties and personal errors such as wall losses (Lee et al., 2006) and the accurate determination of the mass of semi-volatile materials also should be considered (Ehn et al., 2014; McFiggans et al., 2019).

For biogenic SOAs, isoprene and α-pinene SOCs were more abundant in summer, while β-caryophyllene contributed more to SOCs in winter, which may be related to different sources such as biomass burning and other factors such as the volatility of organic compounds (Fig. 13c-d and 14). Similarly, the contributions of isoprene (4.47% versus 3.95%) and α-pinene (2.56% versus 2.44%) derived SOC to OC during daytime and nighttime were apparently elevated in summer (Table S1 and Fig. 13c-d), whereas their contributions in winter were low (Fig. 14a-b). In contrast, the concentrations of β-caryophyllene SOC

were high in winter (average 0.47 and 0.45 μg m$^{-3}$), accounting for 1.89% versus 1.80% of the OC for day- and night-time and being comparable to those (1.95% versus 1.82%) in summer.

The concentrations of anthropogenic SOCs were 2-5 times higher than those of biogenic SOCs in both seasons. The average concentrations of anthropogenic SOCs were 2.12 μg m$^{-3}$ (daytime) and 2.03 μg m$^{-3}$ (nighttime) in winter, higher than the values of 1.0 μg m$^{-3}$ (daytime) and 0.73 μg m$^{-3}$ (nighttime) in summer. Toluene SOC was predominant with the

concentrations of 1.68 ± 0.95 μg m$^{-3}$ versus 1.65 ± 0.85 μg m$^{-3}$ for the day- and night-time in winter, respectively, and with the low levels of 0.82 ± 0.58 μg m$^{-3}$ versus 0.64 ± 1.02 μg m$^{-3}$ in summer. In addition, the levels of naphthalene SOC were 0.44 ± 0.27 μg m$^{-3}$ (daytime) versus 0.38 ± 0.21 μg m$^{-3}$ (nighttime) in winter, and 0.18 ± 0.08 μg m$^{-3}$ (daytime) versus 0.09 ± 0.08 μg m$^{-3}$ (nighttime) in summer. The high anthropogenic SOC concentrations in winter may be attributed to the elevated biomass/biofuel combustion. Although the concentrations of summertime aerosols were lower, the contributions of

anthropogenic SOCs were larger (Table S1 and Fig. 13). Anthropogenic SOCs contributed to 10.1% (daytime) versus 9.12% (nighttime) of total OCs in the aerosols in winter, and 24.7% (daytime) versus 13.6% (nighttime) in summer. Anthropogenic SOCs in summer may be not only related to fossil fuel combustion, but also the increased emissions of plastic emissions in summer (Fujii et al., 2003; Simoneit et al., 2005; Wang et al., 2006; Kong et al., 2013), which warrants further studies.

### 3.4.3 Total contributions of primary OC and SOC in PM$_{2.5}$

Total average concentrations of primary OC were 2.55 ± 1.51 μg m$^{-3}$ (daytime) versus 3.67 ± 1.89 μg m$^{-3}$ (nighttime) in winter and 0.20 ± 0.09 μg m$^{-3}$ (daytime) versus 0.46 ± 0.57 μg m$^{-3}$ (nighttime) in summer, corresponding to 12.4% versus 16.3% of OC for the day- and night-time in winter and 5.24% versus 10.6% of OC in summer, respectively. Additionally, BB-derived OC was detected as the most abundant primary source, followed by fungal spores and plant debris. SOCs including biogenic and anthropogenic sources were estimated as 2.66 ± 1.52 μg m$^{-3}$ (daytime) versus 2.56 ± 1.26 μg m$^{-3}$

(nighttime) in winter and 1.38 ± 0.81 μg m$^{-3}$ (daytime) and 1.09 ± 1.51 μg m$^{-3}$ (nighttime) in summer, accounting for 12.4% versus 11.3% in OC for the day- and night-time in winter and 33.7% versus 21.8% in summer, respectively (Table 1 and Fig 14a-e).

Apparent differences in seasonal characteristics and diurnal variations in organic aerosols were observed between the two seasons in Tianjin. It is worthy to note that the contributions of SOC to OAs in summer were roughly 2 times higher than that in winter, especially for the toluene SOC. Biomass burning OC and β-caryophyllene SOC were more abundant in the winter, while fungal spore- and plant debris-derived OCs, as well as biogenic SOCs contributed more significantly in summer (Fig. 14e). These results are in accordance with previous studies on many megacities in China (Ding et al., 2017) and Beijing (Li et al., 2018). In total, the average contributions of primary and secondary OCs using tracer-based methods discussed above were 24.8% (daytime) versus 27.6% (nighttime) in winter and 38.9% (daytime) versus 32.5% (nighttime) in summer. Besides, there are many other species could contribute to OCs in $PM_{2.5}$ such as aliphatic lipids, dicarboxylic acids, polycyclic aromatic hydrocarbons and other complex compounds (e.g. proteins, amino sugars, and organosulfates) in the ambient atmosphere. GC/MS can only detect small molecules, which account for a small proportion of the total organic matter in the aerosols (Rogge et al., 1993). So, the realization of full analysis of fine particles requires the combination of various analytical techniques such as 2D-GC/MS, HPLC-MS and FT-ICR MS (Nozière et al., 2015), which are summarised in Table S8.

## 4 Conclusions

Atmospheric abundances, molecular compositions, as well as seasonal and diurnal variations in aliphatic lipids (*n*-alkanes, fatty acids, and fatty alcohols), saccharides, biogenic and anthropogenic SOA tracers were investigated in fine aerosols collected at urban Tianjin in the winter of 2016 and the summer of 2017. Results demonstrated that biomass burning was the most abundant source in the winter, while anthropogenic origins among the tracers detected in this study were the predominant contributors to OCs in summer. By comparing the diurnal and seasonal patterns of organic tracers in winter and summer, we found that *n*-alcohols (1310 ng m$^{-3}$ versus 1520 ng m$^{-3}$ for day- and night-time) and *n*-fatty acids (average 666 ng m$^{-3}$ versus 778 ng m$^{-3}$ for day- and night-time) were important organic molecular classes in winter, and the enhanced levels at night may be attributed to the elevated needs for house heating and low boundary layer heights.

Similarly, the dominant species detected in summer were also *n*-alcohols (average 621 ng m$^{-3}$ in the daytime versus 572 ng m$^{-3}$ at night) and fatty acids (410 ng m$^{-3}$ versus 387 ng m$^{-3}$). In contrast to wintertime aerosols, most organic tracers in summer were more abundant in the daytime due to more contributions from marine/biogenic sources by sea breezes when east Asian monsoon prevails in summer. Biogenic SOA tracers from isoprene and α/β-pinene oxidants and fungal spores-derived tracers (arabitol and mannitol) made large contributions to organic aerosols in summer. And the contributions of biogenic SOCs to OCs were in the range of 2.94-16.2% (8.98%) in the daytime and 1.48-22.2% (8.21%) at night, among which 4.47% (daytime) and 3.95% (nighttime) from isoprene, 2.56% (daytime) and 2.44% (nighttime) from α-pinene, 1.95% (daytime) and 1.82% (nighttime) from β-caryophyllene in summer. Fungal spore derived OC contributed to 0.88% and 0.79% of aerosol OC for day- and night-time in summer, respectively. On the other hand, anthropogenic sources were abundant in both seasons. The fractions in summer were 24.7% (daytime) versus 13.6% (nighttime), roughly 2 times more

than wintertime with values as 10.1% and 9.12% for the day- and night-time. Our study highlights that local emissions of primary organic aerosols, biogenic and anthropogenic precursors of secondary organic aerosols, and land/sea breezes and East Asian summer monsoon can affect the atmospheric loadings of organic aerosols in coastal regions of North China.

*Data availability*. The dataset for this paper is available upon request from the corresponding author (fupingqing@tju.edu.cn).

*Competing interests*. The authors declare that they have no conflict of interest.

*Author contributions.* Pingqing Fu and Cong-Qiang Liu designed this research. Laboratory measurements were performed by Yanbing Fan, Linjie Li and Shuang Wang. The manuscript was written by Yanbing Fan and Pingqing Fu with consultation from all other authors.

*Acknowledgements.* This work was supported the National Natural Science Foundation of China (Grant Nos. 41961130384
and 41625014).

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

**Table 1: Abundance of OC from primary sources (BB-derived OC, plant debris-derived OC, fungal spores-derived OC [µg m$^{-3}$] and secondary formation [biogenic SOC and anthropogenic SOC], and their contributions to OC (%) in the samples.**

| | Winter (n=85) | | | | | | | | Summer (n=60) | | | | | | | |
|---|---|---|---|---|---|---|---|---|---|---|---|---|---|---|---|---|
| | Daytime | | | | Nighttime | | | | Daytime | | | | Nighttime | | | |
| | Min | Max | Mean | SD[a] | Min | Max | Mean | SD | Min | Max | Mean | SD | Min | Max | Mean | SD |
| | Abundance (µg m$^{-3}$) | | | | | | | | | | | | | | | |
| biomass burning OC | 0.61 | 7.04 | 2.49 | 1.48 | 0.59 | 7.69 | 3.61 | 1.86 | 0.03 | 0.44 | 0.16 | 0.08 | 0.07 | 2.93 | 0.42 | 0.56 |
| plant debris OC | 0.01 | 0.04 | 0.02 | 0.01 | n.d.[b] | 0.04 | 0.02 | 0.01 | n.d. | 0.02 | 0.01 | 0.00 | n.d. | 0.06 | 0.01 | 0.01 |
| fungal spores OC | 0.01 | 0.13 | 0.04 | 0.03 | 0.01 | 0.10 | 0.04 | 0.02 | n.d. | 0.13 | 0.04 | 0.03 | n.d. | 0.11 | 0.03 | 0.03 |
| sum of primary OC | 0.64 | 7.14 | 2.55 | 1.51 | 0.61 | 7.83 | 3.67 | 1.89 | 0.04 | 0.49 | 0.20 | 0.09 | 0.07 | 2.96 | 0.46 | 0.57 |
| naphthalene SOC | 0.12 | 1.32 | 0.44 | 0.27 | 0.08 | 1.06 | 0.38 | 0.21 | 0.03 | 0.36 | 0.18 | 0.08 | 0.02 | 0.46 | 0.09 | 0.08 |
| toluene SOC | 0.33 | 3.98 | 1.68 | 0.95 | 0.25 | 3.35 | 1.65 | 0.85 | n.d. | 1.92 | 0.82 | 0.58 | n.d. | 5.60 | 0.64 | 1.02 |
| sum of anthropogenic | 0.46 | 5.30 | 2.12 | 1.14 | 0.36 | 3.78 | 2.03 | 0.96 | 0.03 | 2.28 | 1.00 | 0.64 | 0.03 | 6.06 | 0.73 | 1.10 |
| isoprene SOC | 0.01 | 0.08 | 0.03 | 0.02 | 0.01 | 0.07 | 0.03 | 0.02 | 0.01 | 0.13 | 0.05 | 0.03 | 0.02 | 0.29 | 0.06 | 0.06 |
| α-Pinene SOC | 0.02 | 0.18 | 0.06 | 0.03 | n.d. | 0.12 | 0.06 | 0.03 | 0.02 | 0.21 | 0.10 | 0.04 | 0.02 | 0.23 | 0.10 | 0.05 |
| β-caryophyllene SOC | 0.02 | 1.97 | 0.53 | 0.44 | n.d. | 2.03 | 0.55 | 0.43 | n.d. | 0.30 | 0.09 | 0.08 | n.d. | 1.11 | 0.10 | 0.20 |
| sum of biogenic SOC | 0.07 | 2.14 | 0.62 | 0.47 | 0.02 | 2.19 | 0.64 | 0.46 | 0.04 | 0.56 | 0.24 | 0.14 | 0.05 | 1.62 | 0.25 | 0.28 |
| sum of SOC | 0.56 | 7.44 | 2.73 | 1.51 | 0.44 | 5.73 | 2.68 | 1.31 | 0.06 | 2.79 | 1.24 | 0.76 | 0.08 | 7.68 | 0.99 | 1.38 |
| total | 1.39 | 11.2 | 5.28 | 2.79 | 1.08 | 12.2 | 6.34 | 2.84 | 0.12 | 3.03 | 1.45 | 0.82 | 0.16 | 9.31 | 1.45 | 1.76 |
| | Contribution to OC (%) | | | | | | | | | | | | | | | |
| biomass burning OC | 2.74 | 31.1 | 12.1 | 4.93 | 4.64 | 45.5 | 16.0 | 6.88 | 1.05 | 9.03 | 4.14 | 2.18 | 1.89 | 46.7 | 9.62 | 8.73 |
| plant debris OC | 0.01 | 0.26 | 0.10 | 0.05 | 0.03 | 0.20 | 0.08 | 0.03 | 0.06 | 0.44 | 0.22 | 0.08 | 0.06 | 0.45 | 0.22 | 0.08 |
| fungal spores OC | 0.03 | 0.50 | 0.19 | 0.11 | 0.04 | 0.32 | 0.18 | 0.07 | 0.17 | 2.65 | 0.88 | 0.58 | 0.14 | 2.74 | 0.79 | 0.54 |
| sum of primary OC | 2.78 | 31.6 | 12.4 | 5.03 | 4.72 | 45.9 | 16.3 | 6.94 | 1.28 | 9.89 | 5.24 | 2.12 | 2.08 | 47.2 | 10.6 | 8.65 |
| naphthalene SOC | 0.47 | 4.74 | 2.17 | 0.98 | 0.75 | 5.46 | 1.77 | 0.99 | 1.06 | 8.82 | 4.56 | 1.74 | 0.45 | 3.85 | 2.24 | 0.73 |
| toluene SOC | 1.58 | 16.5 | 7.88 | 3.17 | 2.86 | 14.4 | 7.35 | 2.99 | n.d. | 49.2 | 20.1 | 13.1 | n.d. | 27.2 | 11.4 | 8.01 |
| sum of anthropogenic | 2.05 | 21.2 | 10.1 | 3.76 | 3.77 | 19.9 | 9.12 | 3.59 | 1.06 | 54.2 | 24.7 | 14.1 | 0.99 | 30.6 | 13.6 | 8.20 |
| isoprene SOC | 0.03 | 0.64 | 0.19 | 0.13 | 0.04 | 0.49 | 0.16 | 0.09 | 0.53 | 2.27 | 1.32 | 0.54 | 0.65 | 2.72 | 1.48 | 0.61 |
| α-pinene SOC | 0.05 | 0.92 | 0.31 | 0.18 | 0.02 | 0.76 | 0.30 | 0.16 | 0.62 | 4.31 | 2.66 | 0.92 | 0.65 | 4.16 | 2.66 | 0.78 |
| β-caryophyllene SOC | 0.08 | 6.79 | 2.31 | 1.35 | 0.02 | 5.94 | 2.25 | 1.28 | n.d. | 5.26 | 1.93 | 1.47 | n.d. | 4.75 | 1.56 | 1.23 |
| sum of biogenic SOC | 0.27 | 7.80 | 2.80 | 1.46 | 0.07 | 6.67 | 2.70 | 1.38 | 1.52 | 9.65 | 5.91 | 2.17 | 1.30 | 9.37 | 5.70 | 1.79 |
| sum of SOC | 3.47 | 26.5 | 12.9 | 4.78 | 4.71 | 26.5 | 11.8 | 4.50 | 2.58 | 60.9 | 30.6 | 15.7 | 2.29 | 35.5 | 19.3 | 8.85 |
| total | 6.26 | 53.7 | 25.3 | 8.74 | 12.3 | 59.0 | 28.1 | 8.60 | 5.04 | 66.1 | 35.8 | 16.3 | 4.37 | 71.1 | 30.0 | 13.8 |

[a]SD: standard deviation.

[b]n.d.: not detectable. We define those below the limit of detection (LOD) as n.d. The LODs of the target organic compounds in this study were around 0.001-0.08 ng m$^{-3}$.

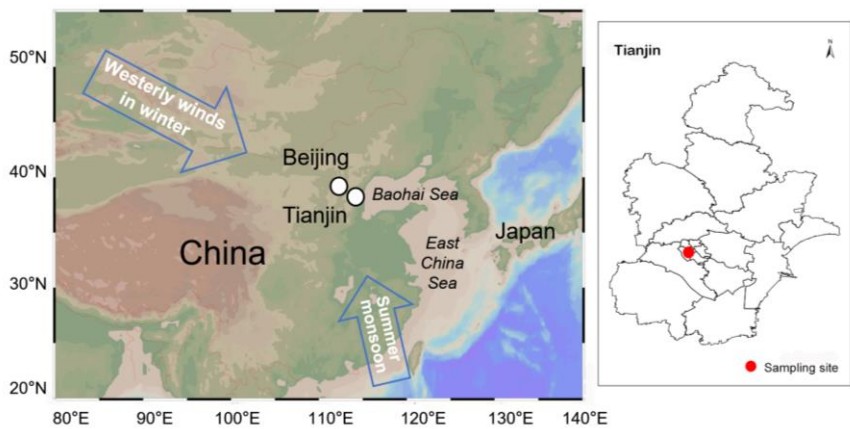

**Figure 1:** Map of showing the location of Tianjin city (left) and the sampling site (right) in Nankai district, Tianjin (the map is from Ocean Data View).

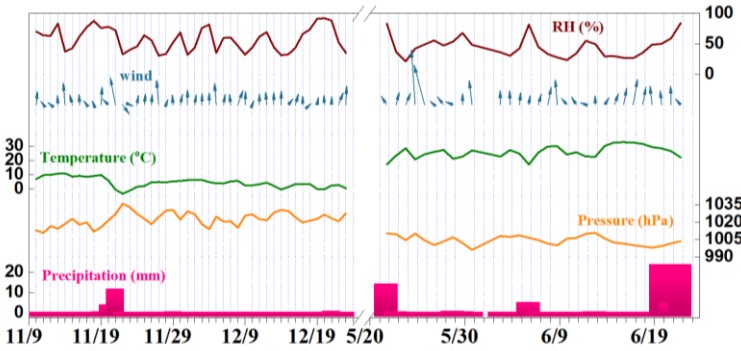

**Figure 2:** Daily variations in relative humidity (RH), wind direction (WD), temperature (T), pressure (P) and precipitation (Precip), The data were obtained from the automatic meteorological station at the sampling sites.

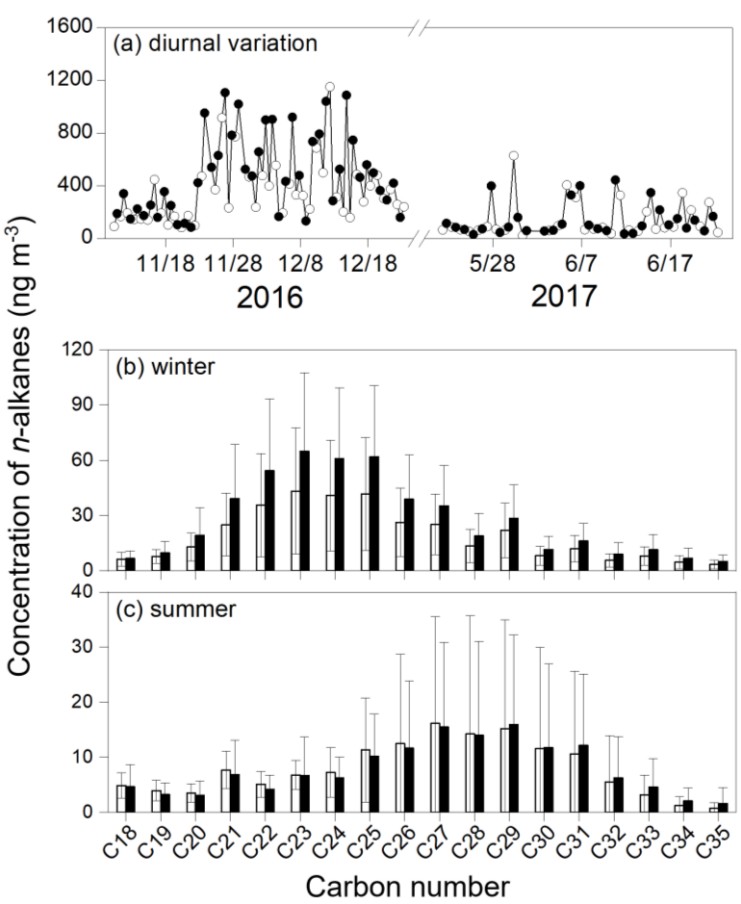

**Figure 3:** Temporal variations (a) and molecular distributions of *n*-alkanes both during the wintertime (b) and summertime (c) in Tianjin. (white and black colour represent day- and night-time, respectively).

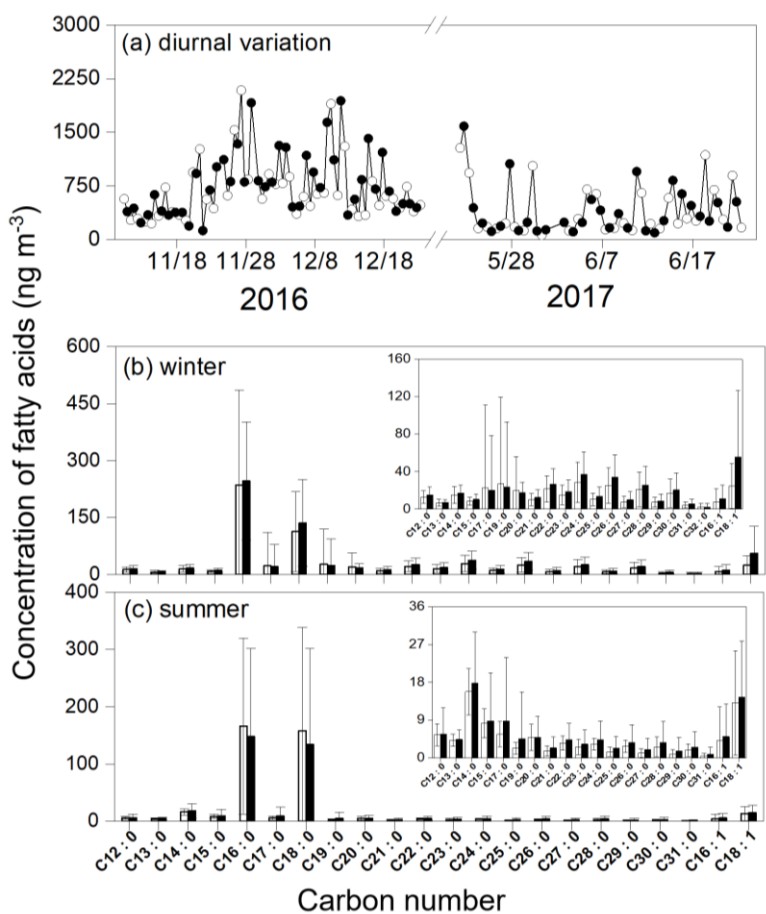

**Figure 4:** Temporal variations (a) and molecular distribution of fatty acids during the wintertime (b) and summertime (c) in Tianjin (white and black colour represent day- and night-time, respectively). The small figures in (b) and (c) are the molecular distributions of fatty acids exclude $C_{16:0}$ and $C_{18:0}$.

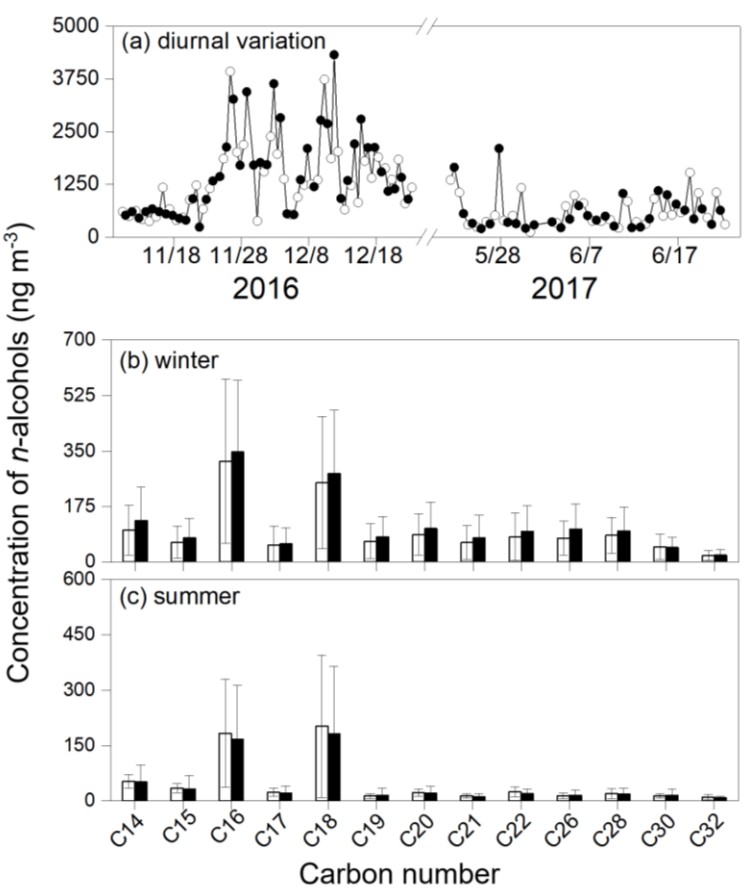

**Figure 5:** Temporal variations (a) and molecular distributions of *n*-alcohols during the wintertime (b) and summertime (c) in Tianjin (white and black colour represent day- and night-time samples, respectively).

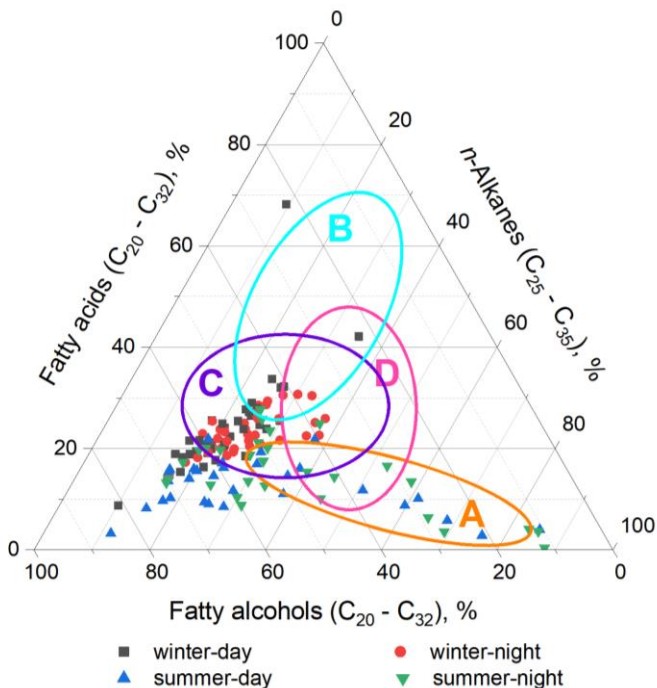

**Figure 6:** Triangular plots of relative abundances of biomarkers detected in Tianjin aerosols during the wintertime and summertime. Three main terrestrial plant waxes including *n*-alkanes (C$_{25}$-C$_{35}$), fatty acids (C$_{20}$-C$_{32}$) and fatty alcohols (C$_{20}$-C$_{32}$) are represented by four shapes of points. The groups A and B are marine aerosols respectively collected from the East China Sea (Kang et al., 2017) in summer and Chichi-Jima Island, the western North Pacific from April 1990 to November 1993 (Kawamura et al., 2003). The category of C represents mountain aerosols from Mt. Tai, China in summer (Fu et al., 2008), and D represents urban aerosols from Beijing in winter (Ren et al., 2016).

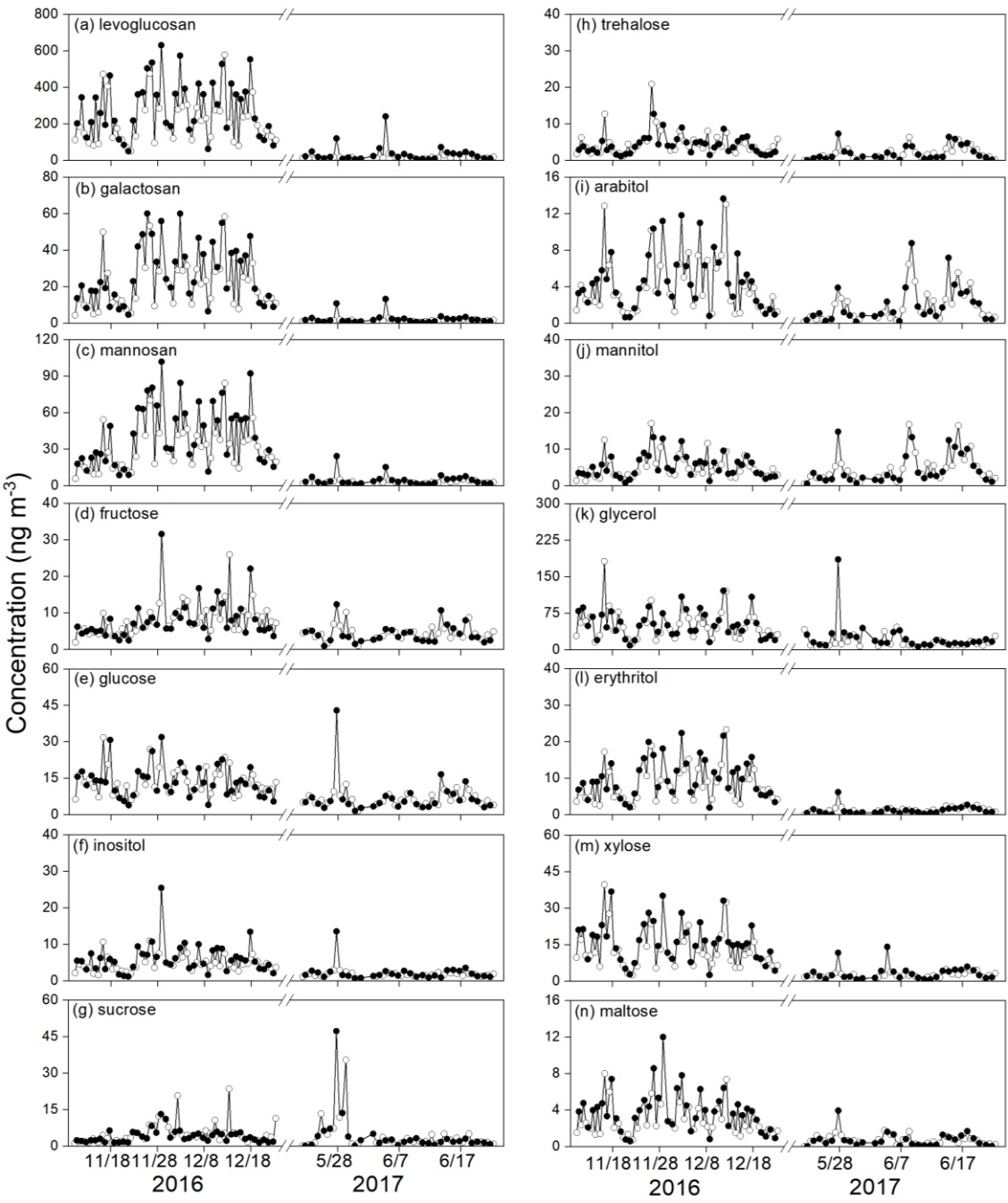

**Figure 7:** Temporal variations in the concentrations of saccharides detected in Tianjin aerosols (white and black colour representing day- and night-time, respectively).

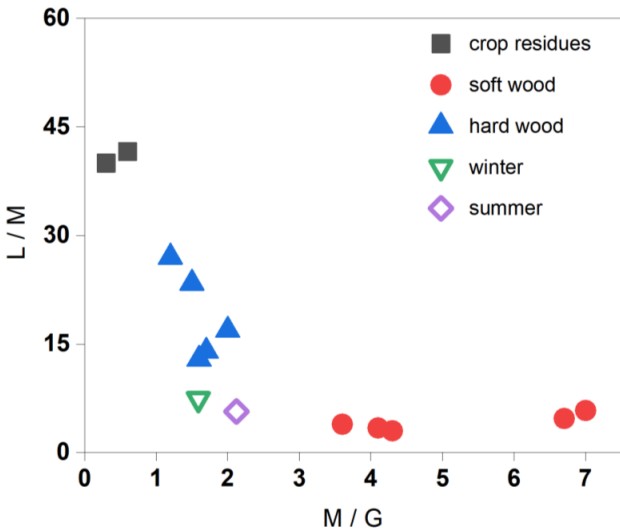

**Figure 8:** Scatter plot of L/M and M/G ratios of samples during winter- and summer-time in this study as well as the ratios of different sources including crop residues, softwood and hardwood from previous literatures (Sheesley et al., 2003; Fine et al., 2004; Oros et al., 2006; Engling et al., 2006).

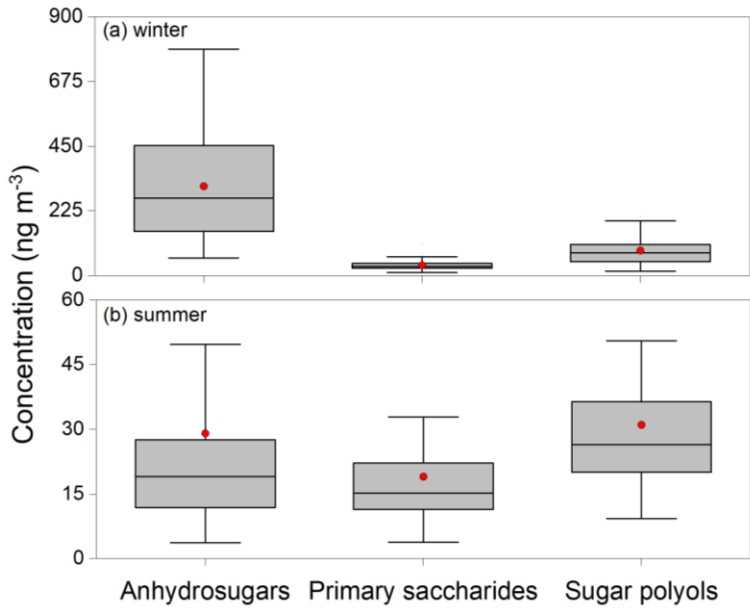

**Figure 9:** Concentrations of sugars including anhydrosugars, primary saccharides and sugar polyols in Tianjin aerosols. Boxes with error bars represent 25th and 75th percentiles of each season. The solid line and red dots in the box represent the median value and average, respectively.

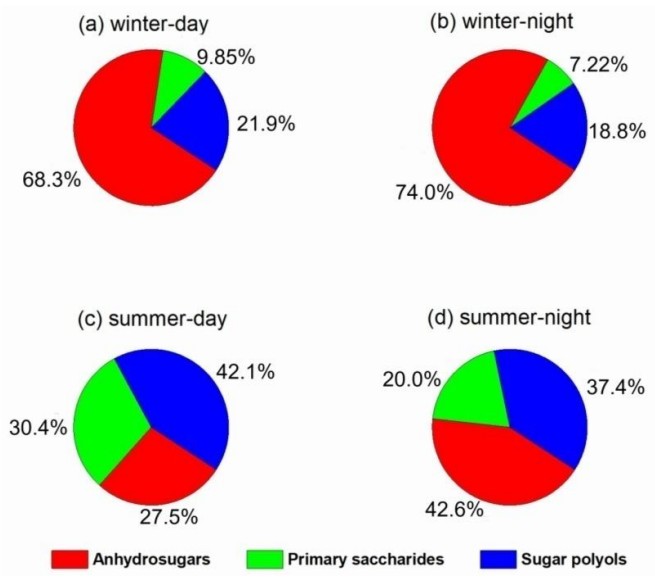

**Figure 10:** Relative contributions of anhydrosugars, primary saccharides and sugar polyols during the day- and night-time in winter and summer.

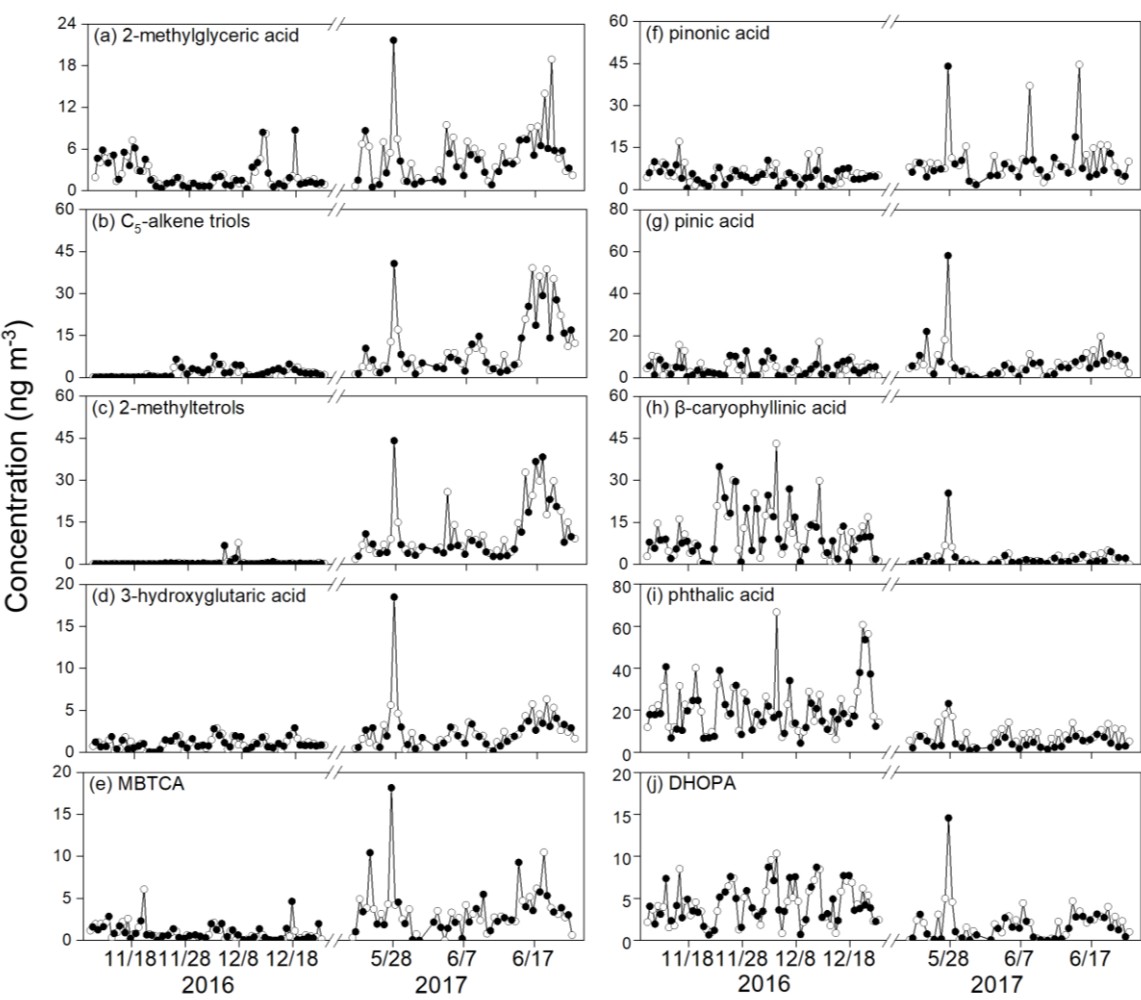

**Figure 11:** Temporal variations in isoprene-, monoterpene-, β-caryophyllene-, naphthalene- and toluene-SOA tracers in PM$_{2.5}$ collected at Tianjin (white and black colour represent day- and night-time, respectively).

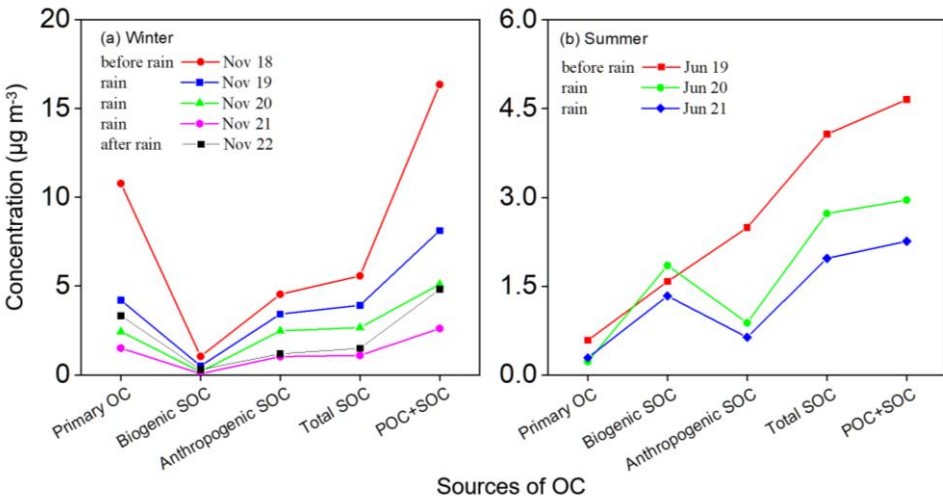

**Figure 12:** The concentration changes of primary and secondary OC during (a) winter- and (b) summer-time on fine and rainy days.

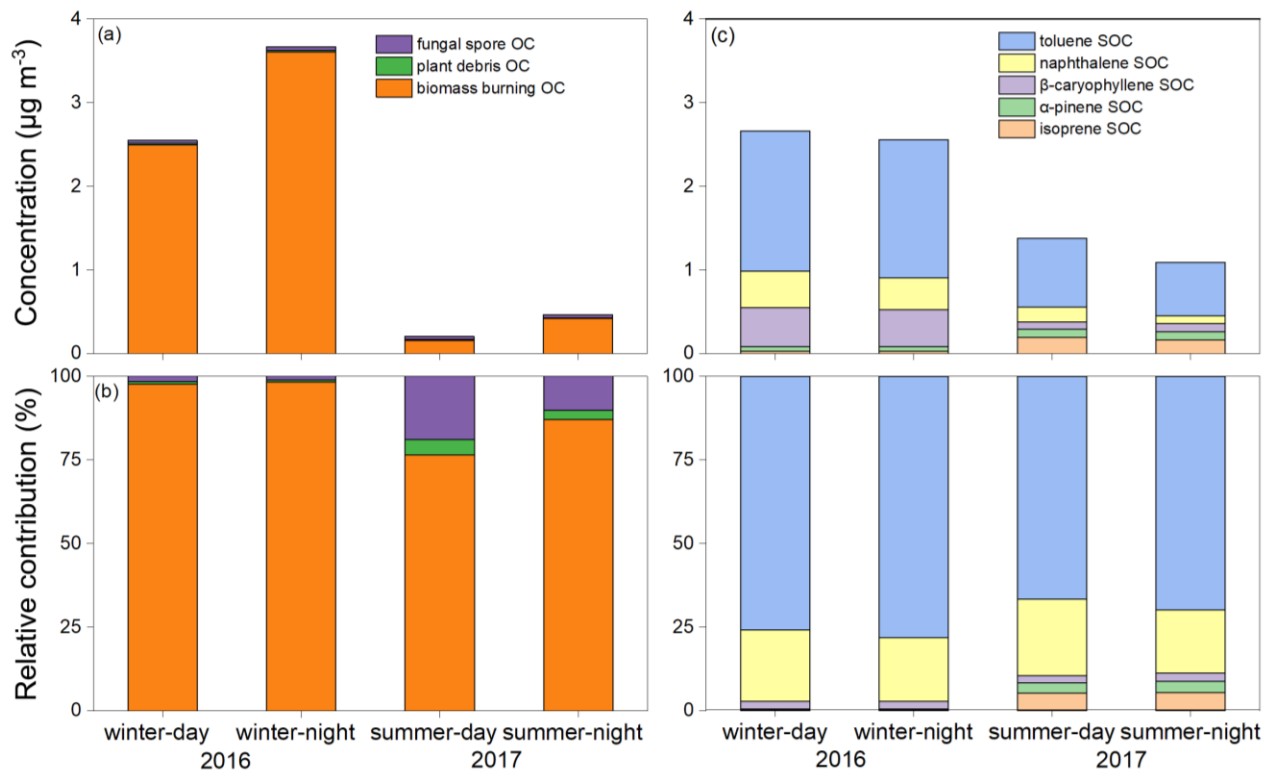

**Figure 23:** (a) Average concentrations of fungal spores-derived OC, plant debris-derived OC and BB-derived OC in $PM_{2.5}$ from Tianjin. (b) Relative contributions of primary OC detected in this study. (c) Average concentrations of toluene SOC, naphthalene SOC, β-caryophyllene SOC, α-pinene SOC and isoprene SOC in $PM_{2.5}$ from Tianjin. (d) Relative contributions of SOC identified in this study.

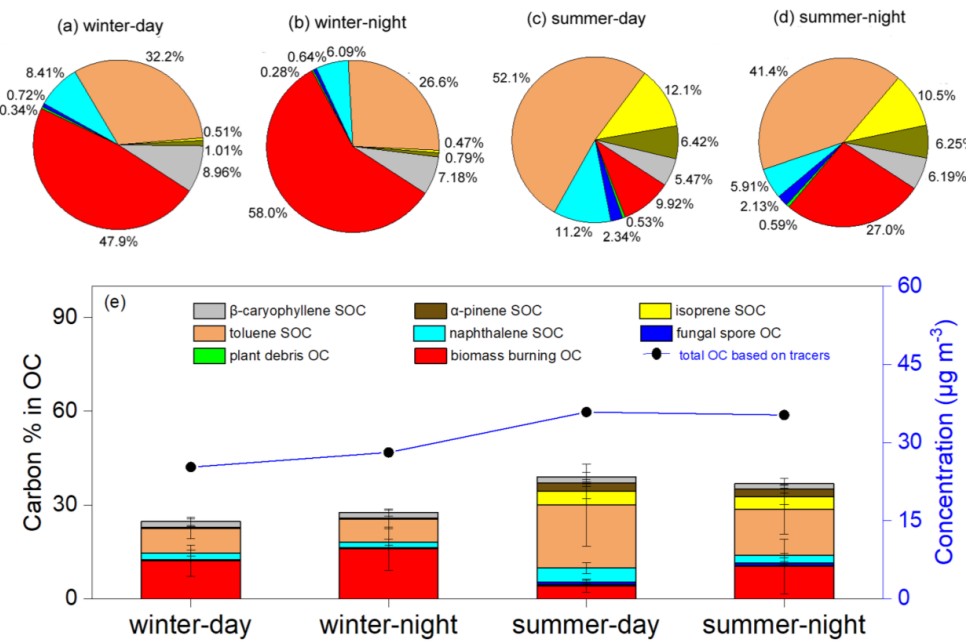

**Figure 14:** Relative contributions of primary and secondary OC (a-d, corresponding to day- and night-time in winter and summer, respectively) and (e) accumulative primary and secondary OC and its contribution to OC in fine particles (%) in Tianjin during winter and summer.