# Peer review of "Large contributions of biogenic and anthropogenic sources to fine organic aerosols in Tianjin, North China"

_Atmospheric Chemistry and Physics, 2019_

## Referee Comment (RC1) · Anonymous Referee #1 · 16 Aug 2019

Comments to Fan et al., Large contributions of biogenic and anthropogenic sources to fine organic aerosols in Tianjin, North China.

The authors collected PM2.5 filter samples diurnally in Tianjian and quantified organic molecular components in two seasons, winter 2016 and early summer 2017. They reported the organic compound levels and estimated the contributions from biomass burning, biogenic emissions and anthropogenic emissions, in view of primary and secondary sources. Although there are quite an amount of studies on source apportionments of PM2.5 in northern China, their reports on aerosol organic compounds are relatively rare and valuable. The writing is easy to follow, while I suggest the authors to

consider the following comments before the paper being published.

Major comments

1. It has been long realized that the major sources of OC in fine aerosols are combustion sources (fossil fuel combustion and biomass burning) and secondary oxidation, in comparison with coarse particles (PM10 or TSP) where dusts and primary biological sources also matter. In the current study, biomass burning, anthropogenic sources (used as toluene and naphthalene SOC), and biogenic sources, as well as plant debris and fungal spore all together are contributing 25-35% of OC (Table 1, Figure 13). The readers may expect if the study could provide more information on the possible sources of the other 65-75%, i.e., the major fraction of OC.

2. It is interesting that saccharides and sugar alcohols showed higher levels in winter than summer in PM2.5 (Fig. 7, Fig. 9). The results imply that sugar alcohols are co-emitted during biomass burning, or co-transport (co-exist) with biomass burning aerosols (Table S3). This observational evidence where speculated in previous studies but not justified. Discussions from such a viewpoint could be interesting to the community.

Minor comments

1. P2, L22, 'due to' change to 'along with'?

2. P2, L33, southerly wind mainly in summer, or throughout the year?

3. Fig. 1, westerly winds prevail mainly in winter, better to add that. It is not necessary to add the location of Mt. Tai. It is better to add an insert figure, showing sampling location in an enlarged map of Tianjin.

4. P4, L29, how is PM2.5 mass measured? Such information is needed to readers. It is interesting to see co-varied PM2.5 and RH. What are the possibly reasons/implications?

5. P5, L1, high concentrations that what level, any threshold?

6. P5, L31-P6, L2, Table S2, it is not easy to understand how WNA (%) is calculated. As they account for only ∼10%, What are the other 90%.

7. P8, L11, 12, unify sugar alcohols and sugar polyols throughout the text.

8. P11, L11, aerosols in Rishiri did not appear in Fu et al. 2010b.

9. P11, L26, give the full name of DHOPA as it appears for the first time.

10. P12, L6, Fig.2, wind direction data could not be seen.

11. P12, L9-13, is there any direct evidence showing burnings in southern of the site in the same period?

12. P12, L17-18, is that true that plastics will evaporate? Please add references.

13. P12, section 3.4.1, it is necessary to add methods to evaluate contributions of BB, fungal spores and plant debris, etc., to OC.

14. P13, L6, 'Table 12a-b, and 13' change to 'Figure 12a-b, and 13'.

15. P13, L6-10, fungal spore OC estimated by Zhu et al. (2016) is based on TSP samples. As these particles are quite large, they have large contributions to OC in their study. It is reasonable that fungal particles have small contributions to PM2.5 samples.

16. P13, L16-17, it is necessary to add reference for the tracer mass fraction factors.

17. P14, L7-9, it would be better to add some reference about the possible plastic emissions if any.

18. Tables S3-4 were not referred in the text?

---

## Referee Comment (RC2) · Anonymous Referee #3 · 6 Nov 2019

Fine aerosol particle (PM2.5) pollution has been one of the most severe environmental problems in the North China Plain (NCP) since the beginning of the new century. In order to elucidate the sources and formation processes of fine particles, here the authors have conducted a field campaign in urban Tianjin, a coastal megacity in NCP, to collect PM2.5 samples on a day/night basis during the winter of 2016 and the summer of 2017. The diurnal patterns are discussed according to the potential effects of land/sea breezes. Tracer-based methods are used to estimate the rough contributions of both primary and secondary sources to aerosol OC. In general, this is an interesting study focusing on the detailed molecular compositions of fine organic aerosols in the coastal regions of China. The results are informative to better understand the diurnal

and seasonal trends of organic aerosols under the influence of local emissions and regional transport. I suggest the manuscript to be accepted for publication in ACP after some revisions based on the comments listed below.

Comments 1. Page 5, lines 2-3: four or five rain events? In addition, please clarify the amount of precipitation in Figure 2; 0.06 mm should be 6 mm? 2. In Page 5, the authors state that during the sampling periods, one rain event occurred in the winter and other four in the summer. Do you have any idea about the source strengths of both primary and secondary OC to total OC between the rainy days and fine days? I'd like to see a bit more discussions on this point in the section of 3.3. 3. In the Caption of Figure 6, It would be better to provide the sampling periods (or seasons) for Categories A – D, which make readers easy to follow. 4. Page 11, Line 19, change "which were five times. . ." to "being five times. . .". 5. Page 12, line 13, "The significant high concentrations. . ." should be "The significantly high concentrations. . .". 6. Page 12, line 18, delete "particularly". 7. Page 13, Section 3.4.2, toluene SOC was found to be the predominant source to OC. What's the main sources of toluene in local regions?

———————————————————

---

## Author Comment (AC1) · 8 Nov 2019

**Responses to comments of the reviewer #1**

We appreciate the helpful comments and suggestions from the reviewer, which greatly improved the quality of our manuscript. The point-to-point responses to the comments are listed below in blue.

**Responses to Reviewer #1:**

Reviewer #1 (Formal Review for Author (shown to author)):
The authors collected $PM_{2.5}$ filter samples diurnally in Tianjian and quantified organic molecular components in two seasons, winter 2016 and early summer 2017. They reported the organic compound levels and estimated the contributions from biomass burning, biogenic emissions and anthropogenic emissions, in view of primary and secondary sources. Although there are quite an amount of studies on source apportionments of $PM_{2.5}$ in northern China, their reports on aerosol organic compounds are relatively rare and valuable. The writing is easy to follow, while I suggest the authors to consider the following comments before the paper being published.

Response: We thank the reviewer for the valuable comments.

Major comments:
1、It has been long realized that the major sources of OC in fine aerosols are combustion sources (fossil fuel combustion and biomass burning) and secondary oxidation, in comparison with coarse particles ($PM_{10}$ or TSP) where dusts and primary biological sources also matter. In the current study, biomass burning, anthropogenic sources (used as toluene and naphthalene SOC), and biogenic sources, as well as plant debris and fungal spore all together are contributing 25-35% of OC (Table 1, Figure 13). The readers may expect if the study could provide more information on the possible sources of the other 65-75%, i.e., the major fraction of OC.

Response: Organic molecular markers detected using GC/MS in this study mainly included aliphatic lipids (*n*-alkanes, fatty acids, *n*-alcohols), saccharides (anhydrosugars, primary saccharides, sugar alcohols), biogenic secondary organic aerosols (BSOA, including isoprene, monoterpene and β-caryophyllene SOAs) and toluene and naphthalene SOAs. We applied mannitol and glucose, isoprene,

monoterpene and β-caryophyllene to evaluate the contributions of primary and secondary biogenic sources to OC, and toluene and naphthalene were used to evaluate the contribution of anthropogenic sources to OC. In total, primary and secondary biogenic OC and anthropogenic OC contributed 25–35% to OC based on these tracers.

At present, there are more than 130 organic molecules that can detected by GC/MS (Fu et al., 2008), such as phthalates, aromatic acids, polycyclic aromatic hydrocarbon, lignin and resin products, hopanes, complex compounds and some possibly existing molecules that are undiscovered now, which can also contributed to OC in $PM_{2.5}$. These contributions of the estimated sources were underestimated based on the tracer method, and some uncertainties should be considered.

In addition, GC/MS is the widely used assay for small organic molecules. It has high temporal resolution and can provide detailed information of organic molecules. And it can quantify the concentrations of organic tracers, such as compounds analyzed in our study. However, GC/MS can only detect small molecules, which account for a small proportion of the total organic matter in the aerosols (Rogge et al., 1993). Other analytical techniques, such as FTICR-MS can detect more than 1000 organic compounds, including macromolecular polymers containing sulfur and nitrogen that are difficult to detect with conventional instruments. So the realization of full analysis of fine particles requires the combination of various analytical techniques (Zhang et al., 2017). Techniques for analyzing organic aerosols are listed in the Table S8. The discussions are shown in the revised manuscript on Pages 16 Lines 10-13 in part of 3.4.3. Table S8 is shown in supporting information.

Table S8. Techniques for analyzing organic aerosols (Ren et al., 2016)

| Analysis targets | Techniques | References |
| --- | --- | --- |
| OC, EC, WSOC | Thermal/optical reflectance, PILS-WSOC | El-Zanan et al., 2012 |
| Organic molecules | HRMS, FTICR-MS | Kourtchev et al., 2016 |
| Small organic molecules | GC/MS, GC/FID, GCXGC/MS, LC/MS, LC/MS/MS, TOF/MS | Kourtchev et al., 2016; Stenson et al., 2002 |
| Organic monomer isotopes | GC-IRMS, EA-MICADAS | Cao et al., 2017 |
| Functional group; Molecules | NMR, FT/IR | Foley et al., 2013; Russell et al., 2011 |
| Online observation | AMS, HR-AMS, CIMS | Chen et al., 2016 |
| Fluorescence | UV-APS, WIBS, EEM-FS | Pöhlker et al., 2012 |

2、It is interesting that saccharides and sugar alcohols showed higher levels in winter than summer in PM$_{2.5}$ (Fig. 7, Fig. 9). The results imply that sugar alcohols are co-emitted during biomass burning, or co-transport (co-exist) with biomass burning aerosols (Table S3). This observational evidence where speculated in previous studies but not justified. Discussions from such a viewpoint could be interesting to the community.

Response: In Figure 7, it is obvious that the concentrations of levoglucosan (a specific tracer of biomass burning), saccharides and sugar alcohols were more abundant in winter than in summer. In addition, saccharides and sugar alcohols had high levels especially when the levoglucosan reached peaks in both seasons, implying that they may share the similar sources to some extent.

Thanks for the reviewer's good suggestion, we provided the fire maps in winter and summer during the sampling periods. Combining with wind direction (WD) to verify the contribution of biomass burning to sugar alcohols. Fire maps during the summer 2017 and winter 2016 in Tianjin are added in Figure S7 in supporting information.

[Figure]

Figure S7: Fire maps during the summer 2017and winter 2016 in Tianjin.

It can be seen clearly that there were large amounts of anthropogenic activities in winter (right in Figure S7). As we all known, the combustion of fossil fuels and biofuels were widely used for house heating in the winter 2016 in China. The biomass burning could contribute to levoglucosan and sugar alcohols. In summer, there were relatively dense fire spots distributed in the North China Plain (NCP) and some south agricultural provinces, such as Jiangsu, Anhui, Henan and Shandong (left in Figure S7). Fu et al. (2008) had reported that there were large-scale burning of wheat straw during May-

June across China. Meanwhile, the wind directions were mainly southerly during winter- and summer-time in Tianjin (Figure 2). Therefore, in addition to the contribution of local biomass burning, the southern wind carried a large number of biomass burning particulate matters from NCP and southern agricultural provinces into Tianjin, which could be the source of levoglucosan and sugar alcohols. Page 10, Lines 8-20.

Minor comments:

1、 P2, L22, 'due to' change to 'along with'?

Response: Following the suggestion, we have revised the sentence. " Tianjin (39°N and 117°E), the largest coastal city of the NCP, located along the Haihe River and being adjacent to the Bohai Sea and East China Sea (Figure 1), has suffered severe haze pollution along with rapid economic and industrial developments during the past decades." Page 2, Line 25.

2、 P2, L33, southerly wind mainly in summer, or throughout the year?

Response: The wind directions have been added, and southerly winds were mainly in winter and summer.

[Figure]

Figure 2: Daily variations in relative humidity (RH), wind direction (WD), temperature (T), pressure (P) and precipitation (Precip), The data were obtained from the automatic meteorological station at the sampling sites.

3、 Fig. 1, westerly winds prevail mainly in winter, better to add that. It is not necessary to add the location of Mt. Tai. It is better to add an insert figure, showing sampling location in an enlarged map of Tianjin.

Response: Thanks. We have provided clear information on the sampling site in Figure 1.

[Figure]

Figure 1: Map of showing the location of Tianjin city (left) and the sampling site (right) in Nankai district, Tianjin.

4、P4, L29, how is PM$_{2.5}$ mass measured? Such information is needed to readers. It is interesting to see co-varied PM$_{2.5}$ and RH. What are the possibly reasons/implications?

Response: Following the suggestion, we have provided clear information on air quality data including AQI, PM$_{2.5}$ and quality grade in Table S7 in supporting information.

The mass concentrations of PM$_{2.5}$ were available from the website of China air quality online monitoring and analysis platform. And we have added the source of PM$_{2.5}$ on Page 5, Lines 1-3.

5、P5, L1, high concentrations that what level, any threshold?

Response: High concentrations represent the peaks relative to the sample concentrations in the surrounding days (see Page 5, Line 17).

6、P5, L31-P6, L2, Table S2, it is not easy to understand how WNA (%) is calculated. As they account for only ∼10%, What are the other 90%.

Response: Plant wax carbon number is expressed by the difference between a certain carbon number concentration and the average of its adjacent two carbon number concentrations: Wax $C_n = C_n-[(C_{n+1}+C_{n-1})/2]$, when the $C_n < 0$, $C_n = 0$; (n represents odd carbon number)

The % of Wax $C_n$ is the contribution of biogenic *n*-alkane that is derived from high plant waxes (Ren et al., 2016). The certain carbon we used to calculate contribution of plant wax were odd carbons from $C_{25}$ to $C_{33}$ (Kang et al., 2017; Ren et al., 2016). (%Wax $C_n$ = Wax $C_n$/$\Sigma C_n$ ×100%)

In this study, plant wax *n*-alkanes only accounted for ~10%, and the large sources of *n*-alkanes maybe attributed to anthropogenic activities, such as biomass burning and fossil fuels, as mentioned in this study. Furthermore, there may also be some other sources that we need to further find afterwards (Page 6, Lines 15). The computational formula is provided in Table S2.

7、P8, L11, 12, unify sugar alcohols and sugar polyols throughout the text.

Response: Thanks. We have corrected it as suggested in the revised manuscript.

8、P11, L11, aerosols in Rishiri did not appear in Fu et al. 2010b.

Response: The data in Rishiri aerosols have not been unpublished, so we removed this information in the revision.

9、P11, L26, give the full name of DHOPA as it appears for the first time.

Response: Thanks for the reviewer for pointing this mistake. We have corrected it on Page 12, Line 26.

10、P12, L6, Fig.2, wind direction data could not be seen.

Response: We have corrected it in Figure 2.

11、P12, L9-13, is there any direct evidence showing burnings in southern of the site in the same period?

Response: Yes. We've provided a fire spot map in Figure S7 in supporting information to show direct evidence of burnings in southern of the site.

12、P12, L17-18, is that true that plastics will evaporate? Please add references.

Response: We have modified the sentence and added some references as follows: "…elevated temperature promotes the evaporation of phthalate esters from plastic products (Fujii et al., 2003; Wang et al., 2007)."

13、P12, section 3.4.1, it is necessary to add methods to evaluate contributions of BB,

fungal spores and plant debris, etc., to OC.

Response: Thanks. The calculated methods for contributions of POC and SOC to OC were shown in the discussion. We have put this content in the method part (see Page 4, Lines 21-26).

14、P13, L6, 'Table 12a-b, and 13' change to 'Figure 12a-b, and 13'.

Response: Thanks. We have corrected it as suggested in the revised manuscript (see Page 14, Line 19).

15、P13, L6-10, fungal spore OC estimated by Zhu et al. (2016) is based on TSP samples. As these particles are quite large, they have large contributions to OC in their study. It is reasonable that fungal particles have small contributions to $PM_{2.5}$ samples.

Response: We agree that fungal spores and plant debris have large particle sizes and usually exist in coarse particles, while only some small fragments exist in fine particles with low content.

16、P13, L16-17, it is necessary to add reference for the tracer mass fraction factors.

Response: We have added it as suggested on Page 4, Line 25.

17、P14, L7-9, it would be better to add some reference about the possible plastic emissions if any.

Response: We have added references as suggested on Page 15, Line 23.

18、Tables S3-4 were not referred in the text?

Response: Thanks. We have added it on Page 10, Lines 2-4.

References:

Fu, P. Q., Kawamura, K., Kanaya, Y., and Wang, Z. F.: Contributions of biogenic volatile organic compounds to the formation of secondary organic aerosols over Mt. Tai, Central East China, Atmos. Environ., 44, 4817-4826, 2010a.(Sindelarova et al., 2014)

Fu, P. Q., Kawamura, K., Okuzawa, K., Aggarwal, S. G., Wang, G. H., Kanaya, Y., and Wang, Z. F.: Organic molecular compositions and temporal variations of summertime mountain aerosols over Mt. Tai, North China Plain, J. Geophys. Res., [Atmos], 113, D19107, doi:10.1029/2008JD009900, 2008.

FUJII, SHINOHARA, LIM, OTAKE, and KUMAGAI: A study on emission of phthalate esters from plastic materials using a passive flux sampler, Atmospheric Environment, 37, 5495-5504, 2003.

Kong, S., Ji, Y., Liu, L., Chen, L., and Zhao, X.: Spatial and temporal variation of phthalic acid esters (PAEs) in atmospheric PM10 and PM2.5 and the influence of ambient temperature in Tianjin, China, Atmospheric Environment, 74, 199-208, 2013.

Kleindienst, T. E., Jaoui, M., Lewandowski, M., Offenberg, J. H., Lewis, C. W., Bhave, P. V., and Edney, E. O.: Estimates of the contributions of biogenic and anthropogenic hydrocarbons to secondary organic aerosol at a southeastern US location, Atmos. Environ., 41, 8288-8300, 2007.

Qiao, B., Chen, Y., Tian, M., Wang, H., Yang, F., Shi, G., Zhang, L., Peng, C., Luo, Q., and Ding, S.: Characterization of water soluble inorganic ions and their evolution processes during PM2.5 pollution episodes in a small city in southwest China, Science of The Total Environment.

Rogge, W. F., Hlldemann, L. M., Mazurek, M. A., and R., a. C. G.: Sources of Fine Organic Aerosol. 2. Noncatalyst and Catalyst-Equipped Automobiles and Heavy-Duty Diesel Trucks, Environ. Sci. Technol., 27, 636-651, 1993.

Ren, L. J., Fu, P. Q., He, Y., Hou, J. Z., Chen, J., Pavuluri, C. M., Sun, Y. L., and Wang, Z. F.: Molecular distributions and compound-specific stable carbon isotopic compositions of lipids in wintertime aerosols from Beijing, Scientific Reports, 6, 10.1038/srep27481, 2016.

Simoneit, B. R. T., Medeiros, P. M., and Didyk, B. M.: Combustion products of plastics as indicators for refuse burning in the atmosphere, Environ. Sci. Technol., 39, 6961-6970, 2005.

Sindelarova, K., Granier, C., Bouarar, I., Guenther, A., Tilmes, S., Stavrakou, T., Müller, J.-F., Kuhn, U., Stefani, P., and Knorr, W.: Global data set of biogenic VOC emissions calculated by the MEGAN model over the last 30 years, Atmospheric Chemistry & Physics, 14, 10725-10788, 2014.

Tao, J., Zhang, L. M., Cao, J. J., and Zhang, R. J.: A review of current knowledge concerning PM2.5 chemical composition, aerosol optical properties and their relationships across China, Atmospheric Chemistry and Physics, 17, 9485-9518, 10.5194/acp-17-9485-2017, 2017.

Wang, Kawamura K , and Zhao X , e. a.: Identification, abundance and seasonal variation of anthropogenic organic aerosols from a mega-city in China, Atmospheric Environment, 41, 407-416, 2007.

Xu, W. Q., Han, T. T., Du, W., Wang, Q. Q., Chen, C., Zhao, J., Zhang, Y. J., Li, J., Fu, P. Q., Wang, Z. F., Worsnop, D. R., and Sun, Y. L.: Effects of Aqueous-Phase and Photochemical Processing on Secondary Organic Aerosol Formation and Evolution in

Beijing, China, Environmental Science & Technology, 51, 762-770, 10.1021/acs.est.6b04498, 2017.

Zhang, Y. L., Ren, H., Sun, Y. L., Cao, F., Chang, Y. H., Liu, S. D., Lee, X. H., Agrios, K., Kawamura, K., Liu, D., Ren, L. J., Du, W., Wang, Z. F., Prevot, A. S. H., Szida, S., and Fu, P. Q.: High Contribution of Nonfossil Sources to Submicrometer Organic Aerosols in Beijing, China, Environmental Science & Technology, 51, 7842-7852, 10.1021/acs.est.7b01517, 2017.

Zheng, G. J., Duan, F. K., Su, H., Ma, Y. L., Cheng, Y., Zheng, B., Zhang, Q., Huang, T., Kimoto, T., Chang, D., Poschl, U., Cheng, Y. F., and He, K. B.: Exploring the severe winter haze in Beijing: the impact of synoptic weather, regional transport and heterogeneous reactions, Atmospheric Chemistry and Physics, 15, 2969-2983, 10.5194/acp-15-2969-2015, 2015.

---

## Author Comment (AC2) · 8 Nov 2019

**Responses to comments of the reviewer #2**

We appreciate the helpful comments and suggestions from the reviewer, which greatly improved the quality of our manuscript. The point-to-point responses to the comments are listed below in blue.

**Responses to Reviewer #2:**

Fine aerosol particle ($PM_{2.5}$) pollution has been one of the most severe environmental problems in the North China Plain (NCP) since the beginning of the new century. In order to elucidate the sources and formation processes of fine particles, here the authors have conducted a field campaign in urban Tianjin, a coastal megacity in NCP, to collect $PM_{2.5}$ samples on a day/night basis during the winter of 2016 and the summer of 2017. The diurnal patterns are discussed according to the potential effects of land/sea breezes. Tracer-based methods are used to estimate the rough contributions of both primary and secondary sources to aerosol OC. In general, this is an interesting study focusing on the detailed molecular compositions of fine organic aerosols in the coastal regions of China. The results are informative to better understand the diurnal and seasonal trends of organic aerosols under the influence of local emissions and regional transport. I suggest the manuscript to be accepted for publication in ACP after some revisions based on the comments listed below.

Comments:

1. Page 5, lines 2-3: four or five rain events? In addition, please clarify the amount of precipitation in Figure 2; 0.06 mm should be 6 mm?

Response: Thanks for the reviewer for pointing this mistake. We have corrected the amount of precipitation of the four rain events in Figure 2. They were occurred on November 20-22 in 2016, May 21-22 and June 5-6, 20-22 in 2017 during sampling periods (please see Figure 2 and Page 5, Line 18-20).

[Figure]

Figure 2. (revised)

2. In Page 5, the authors state that during the sampling periods, one rain event occurred in the winter and other four in the summer. Do you have any idea about the source strengths of both primary and secondary OC to total OC between the rainy days and fine days? I'd like to see a bit more discussions on this point in the section of 3.3.

Response: Thanks for the reviewer's suggestions. We have added the following paragraph in the revised manuscript (see Page 13, Lines 20-30).
"In this study, four rain events were recorded during the sampling periods. It is interesting to note that there were obvious differences between winter- and summer-time samples in terms of the contributions of primary and secondary OC to total OC on rainy and fine days (Figure 12). The concentrations of primary and secondary OC decreased dramatically on rainy days in both seasons, mainly due to the washout effect on pollutants. In winter, the levels of primary OC were higher than secondary OC (mainly from anthropogenic VOCs) before the rain events. Although the concentrations of primary and secondary OC decreased on rainy days, the level of primary OC had a substantial reduction (Figure 12a). However, in summer, the concentrations of secondary OC (both biogenic and anthropogenic SOC) were significantly higher than primary OC before the rain event. We found that the summertime rain event affected little on the levels of primary OC and biogenic SOC, but it decreased the anthropogenic SOC obviously. Such seasonal differences may be attributed to the important and persistent sources such as fossil fuel combustion and biomass burning in the local regions in winter and biogenic VOC emissions in summer."

[Figure]

Figure 12: The concentration changes of primary and secondary OC during (a) winter- and (b) summer-time on fine and rainy days.

3. In the Caption of Figure 6, It would be better to provide the sampling periods (or seasons) for Categories A – D, which make readers easy to follow.

Response: Thanks. We have corrected it as suggested in the caption of Figure 6 on page 30 of the revised manuscript.

4. Page 11, Line 19, change "which were five times…" to "being five times…".

Response: Thanks for the reviewer's carefulness. We have corrected it as suggested on page 12, Line 19 of the revised manuscript.

5. Page 12, line 13, "The significant high concentrations…" should be "The significantly high concentrations…".

Response: Thanks for the reviewer for pointing this mistake. We have corrected it as suggested on page 13, Line 14 of the revised manuscript.

6. Page 12, line 18, delete "particularly".

Response: We have deleted it on page 13, Line 19 of the revised manuscript.

7. Page 13, Section 3.4.2, toluene SOC was found to be the predominant source to OC. What's the main sources of toluene in local regions?

Response: Thanks for the reviewer's suggestion. Vehicle emissions may be the main sources for toluene in Tianjin, and solvent usage in electronics manufacturing and

household product industries maybe also the sources of toluene in some other regions. And we have added this point on page 14, Line 31-32 and page 15, Line 1.

**References:**

Fu, P. Q., Aggarwal, S. G., Chen, J., Li, J., Sun, Y. L., Wang, Z. F., Chen, H. S., Liao, H., Ding, A. J., Umarji, G. S., Patil, R. S., Chen, Q., and Kawamura, K.: Molecular markers of secondary organic aerosol in Mumbai, India, Environ. Sci. Technol., 50, 4659-4667, 10.1021/acs.est.6b00372, 2016.

Kleindienst, T. E., Jaoui, M., Lewandowski, M., Offenberg, J. H., Lewis, C. W., Bhave, P. V., and Edney, E. O.: Estimates of the contributions of biogenic and anthropogenic hydrocarbons to secondary organic aerosol at a southeastern US location, Atmos. Environ., 41, 8288-8300, 2007.

Wang, Kawamura K , and Zhao X , e. a.: Identification, abundance and seasonal variation of anthropogenic organic aerosols from a mega-city in China, Atmospheric Environment, 41, 407-416, 2007.

Widiana, D. R., Wang, Y. C., You, S. J., and Wang, Y. F.: Source apportionment and health risk assessment of ambient volatile organic compounds in primary schools in Northern Taiwan, Int. J. Environ. Sci. Technol., 16, 6175-6188, 10.1007/s13762-018-2157-1, 2019.